# Dopamine in the dorsal bed nucleus of stria terminalis signals Pavlovian sign-tracking and reward violations

Utsav Gyawali[1,2]*, David A Martin[2], Fangmiao Sun[3], Yulong Li[3], Donna Calu[1,2]*

[1]Program in Neuroscience, University of Maryland School of Medicine, Baltimore, United States; [2]Department of Anatomy and Neurobiology, University of Maryland School of Medicine, Baltimore, United States; [3]State Key Laboratory of Membrane Biology, Peking University School of Life Sciences; PKU-IDG/McGovern Institute for Brain Research; Peking-Tsinghua Center for Life Sciences, Beijing, China

**\*For correspondence:**
ugyawali@umaryland.edu (UG);
dcalu@som.umaryland.edu (DC)

**Competing interest:** The authors declare that no competing interests exist.

**Abstract** Midbrain and striatal dopamine signals have been extremely well characterized over the past several decades, yet novel dopamine signals and functions in reward learning and motivation continue to emerge. A similar characterization of real-time sub-second dopamine signals in areas outside of the striatum has been limited. Recent advances in fluorescent sensor technology and fiber photometry permit the measurement of dopamine binding correlates, which can divulge basic functions of dopamine signaling in non-striatal dopamine terminal regions, like the dorsal bed nucleus of the stria terminalis (dBNST). Here, we record GRAB$_{DA}$ signals in the dBNST during a Pavlovian lever autoshaping task. We observe greater Pavlovian cue-evoked dBNST GRAB$_{DA}$ signals in sign-tracking (ST) compared to goal-tracking/intermediate (GT/INT) rats and the magnitude of cue-evoked dBNST GRAB$_{DA}$ signals decreases immediately following reinforcer-specific satiety. When we deliver unexpected rewards or omit expected rewards, we find that dBNST dopamine signals encode bidirectional reward prediction errors in GT/INT rats, but only positive prediction errors in ST rats. Since sign- and goal-tracking approach strategies are associated with distinct drug relapse vulnerabilities, we examined the effects of experimenter-administered fentanyl on dBNST dopamine associative encoding. Systemic fentanyl injections do not disrupt cue discrimination but generally potentiate dBNST dopamine signals. These results reveal multiple dBNST dopamine correlates of learning and motivation that depend on the Pavlovian approach strategy employed.

## Editor's evaluation

Gyawali et al. report individual differences in extended amygdala dopamine signaling of natural and drug reward associated cues. The authors provide compelling evidence of dopamine correlates of Pavlovian natural reward and instrumental drug reward associations in rats, and their results are of broad interest to those studying brain reward systems with significance for cue-induced relapse vulnerability, in particular.

## Introduction

Survival depends on learning to associate environmental cues with food or other natural rewards. Individual differences in learning and motivational processes support the acquisition, expression, and updating of cue-reward associations. Recent evidence suggests that distinct learning strategies are predictive of dysregulated motivation for drug-associated cues/conditioned stimuli (CS) (*Chang et al., 2022*; *Martin et al., 2022*; *Pitchers et al., 2017*; *Saunders et al., 2013*). Midbrain and striatal

dopamine signals are broadly implicated in a diverse array of learning and motivational processes, including CS-reward associations underscoring the importance of dopamine (DA) in adaptive behavior that promotes survival (*Langdon et al., 2018*; *Lee et al., 2022*; *Nasser et al., 2017*). Yet considerably less is known about the role of DA signals in areas outside of the striatum during adaptive and maladaptive cue-reward learning. Recent advances in fluorescent sensor technology and fiber photometry permit the measurement of DA binding correlates (*Labouesse et al., 2020*). These new techniques can reveal understudied functions of DA signaling in non-striatal DA terminal regions like the dBNST, an extended amygdala nucleus, that is critical for dysregulated CS-triggered opioid relapse (*Gyawali et al., 2020*). Here, we characterize basic dBNST DA correlates by recording fluorescent $GRAB_{DA}$ signals during a Pavlovian task that distinguishes two distinct relapse vulnerability phenotypes.

Recent studies identify unique learning strategies that predict heightened CS-triggered relapse vulnerability (*Chang et al., 2022*; *Martin et al., 2022*; *Pitchers et al., 2017*; *Saunders et al., 2013*). In particular, a simple Pavlovian Lever Autoshaping task distinguishes two extreme tracking phenotypes: (1) sign-tracking (ST) rats that approach and vigorously engage with the reward predictive lever cue, even though cue interaction is not necessary to obtain food reward and (2) goal-tracking rats that interact with the food cup during cue presentation where food reward is delivered after lever retraction (*Boakes, 1977*; *Flagel et al., 2007*; *Hearst and Jenkins, 1974*; *Meyer et al., 2012*). Sign-tracking rats show heightened CS-triggered drug relapse vulnerability compared to goal-trackers. A third group called intermediates approach both the food cup and lever at similar levels, and their relapse vulnerability is like that of goal-tracking rats (*Saunders and Robinson, 2010*). Fast scan cyclic voltammetry recording of real-time dopamine indicated that sign-, but not goal-tracking, evokes increases in phasic fluctuations in DA in the nucleus accumbens (NAc) during CS presentation (*Flagel et al., 2011*). NAc DA is necessary for both the expression of sign-tracking and for sign-trackers' heightened CS-triggered drug relapse, but not for goal-trackers or their relapse behavior (*Saunders et al., 2013*). Given the critical role of dBNST in CS-triggered relapse, we aimed to determine whether there are similar individual differences in dBNST DA signaling in sign- and goal-tracking rats using the Pavlovian Lever Autoshaping task (*Buffalari and See, 2009*; *Gyawali et al., 2020*; *Silberman and Winder, 2013b*).

Midbrain dopamine neuron activity strengthens cue-outcome associations by serving as a bidirectional prediction error signal where unexpected reward delivery increases and omitted reward decreases dopamine neuron firing relative to expected reward (*Montague et al., 1996*; *Schultz, 2015*; *Schultz et al., 1997*). Over the course of learning, the phasic dopamine activity transfers from the unconditioned stimulus (US) to the CS (*Montague et al., 1996*; *Schultz, 2015*; *Schultz et al., 1997*). In the NAc, the transfer of dopamine signals from the US to the CS occurs more robustly in ST compared to GT rats (*Flagel et al., 2011*; *Lee et al., 2018*; *Saddoris et al., 2016*), and NAc DA antagonism reduces sign-tracking but not goal-tracking behaviors (*Saunders and Robinson, 2012*). Together, these studies support the Pavlovian lever autoshaping task (and sign-tracking) as a reliable framework for studying dopamine's role in regions of the brain critically involved in cue-motivated natural and drug reward-seeking behaviors.

The dBNST receives dense dopaminergic input from several midbrain regions including the ventral tegmental area, ventral periaqueductal gray, and to a much lesser extent, the substantia nigra (*Hasue and Shammah-Lagnado, 2002*; *Meloni et al., 2006*). dBNST dopamine is associated with a variety of reward-motivated behaviors. dBNST dopamine release is increased during intra-oral sucrose infusion and in response to cues that predict intracranial self-stimulation of the medial forebrain bundle (*Lin et al., 2020*; *Park et al., 2012*; *Park et al., 2013*). Furthermore, dopamine antagonist injections in the dBNST reduce responding to sucrose in a binge eating paradigm (*Maracle et al., 2019*). All major drugs of abuse, including opioids, increase extracellular dopamine in the BNST and dBNST dopamine antagonism reduces cocaine self-administration and ethanol seeking (*Carboni et al., 2000*; *Eiler et al., 2003*; *Epping-Jordan et al., 1998*). Despite these studies implicating dBNST dopamine in motivated behaviors, a comprehensive characterization of endogenous dBNST dopamine dynamics in cue-induced behaviors is lacking. To address this, we used a dopamine sensor $GRAB_{DA}$ in combination with fiber photometry to examine the basic properties of the dBNST dopamine signals; their role during lever autoshaping, reward violations, outcome specific-satiety and during systemic fentanyl administration (*Sun et al., 2018*).

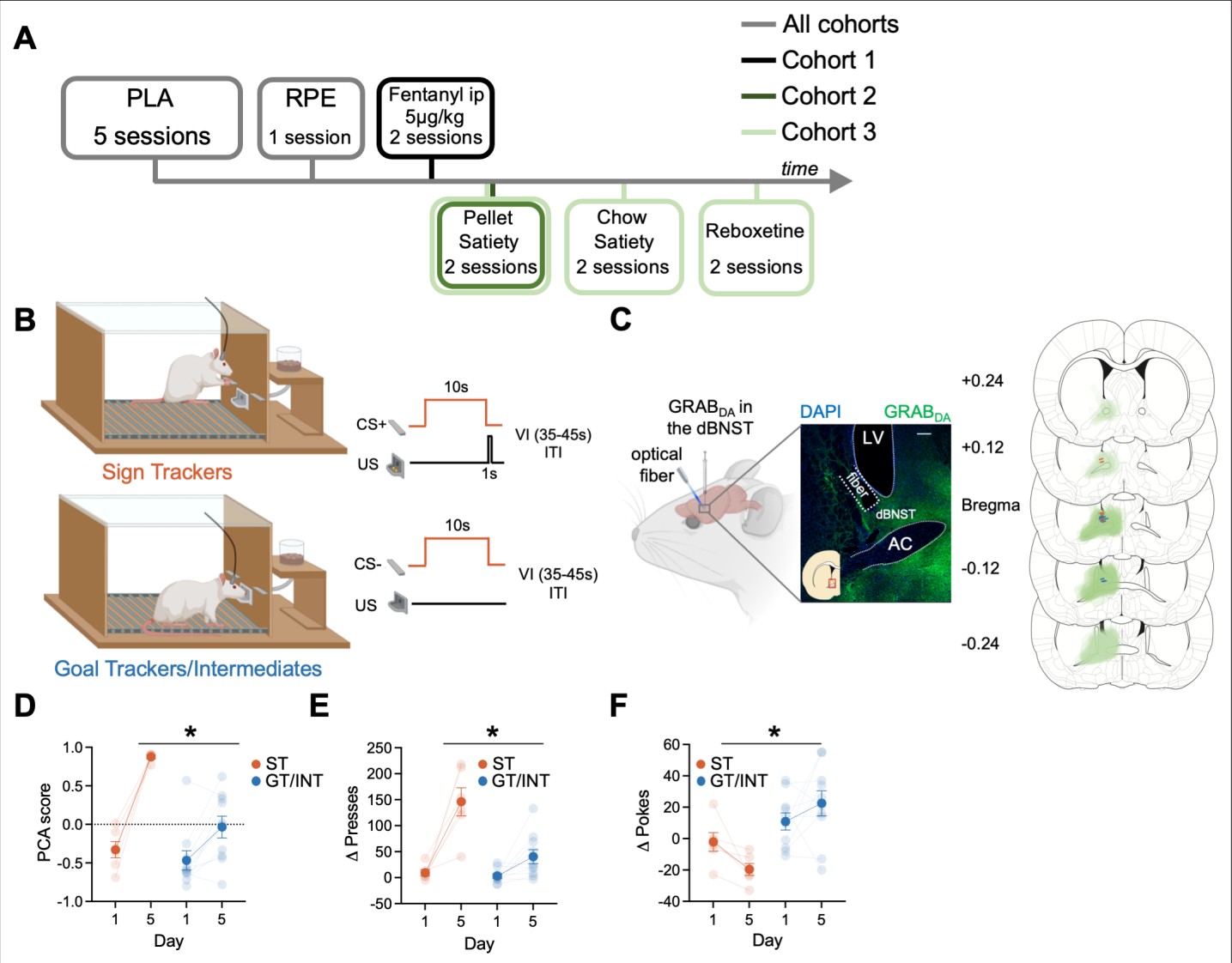

**Figure 1.** Individual differences emerge during Pavlovian lever autoshaping (PLA). (**A**) Experimental timeline. We trained all rats for five daily reinforced PLA sessions to determine their tracking groups followed by a single reward prediction error (RPE) session. We injected the first cohort of rats with i.p fentanyl in PLA and tested the second and the third cohort of rats on two counterbalanced PLA pellet satiety sessions. We tested the third cohort of rats on two counterbalanced PLA chow satiety and with reboxetine i.p. injection sessions. (**B**) PLA sessions consisted of the presentation of 10 s of cue (either conditioned stimulus, CS + or CS− lever, pseudorandom order with an intertrial interval (ITI) varying (variable interval (VI)) between 35 and 45 s) followed by lever retraction and delivery of two food pellets in the food cup. Some rats (Sign Trackers, STs) engage with the cue while others (Goal trackers, GTs) wait in the food cup during the cue period. Others display both lever and food cup behaviors (Intermediates, INTs) (**C**) Left: representative expression of GRAB$_{DA}$ construct and fiber placement in dorsal bed nucleus of stria terminalis (dBNST). White scale bar: 250 μm. Right: The extent of GRAB$_{DA}$ expression and fiber placement across five coronal planes with anterior distance from bregma (millimeters) in the dBNST in STs (orange) and GT/INTs (blue). Drawings were adapted from Figures 31, 32, 33, 34, and 35 from *Paxinos and Watson, 2006*. (**D**) Average Pavlovian conditioned approach (PCA) scores for STs and GT/INTs on Day 1 and Day 5 of PLA. (**E**) Average Δ Presses (CS+) – (CS−) on Day 1 and Day 5. (**F**) Average Δ Pokes (CS+) – (CS−) on Day 1 and Day 5. Data are mean ± SEM. *p<0.05.

## Results

### BNST GRAB$_{DA}$ signals respond to cues and rewards in a Pavlovian learning task

We sought to determine if BNST GRAB$_{DA}$ signals correlate with individual differences in approach to Pavlovian cues (Experiment timeline in *Figure 1A*). First, we trained rats in Pavlovian Lever Approach (PLA) for five days (*Figure 1B*) to examine the acquisition of lever- and food cup-directed behaviors

across training in sign and goal tracking/intermediate rats. Representative and histological inventory of GRAB_DA expression from these rats is shown in *Figure 1C*. We analyzed the behavioral PCA score using a mixed ANOVA with between-subject factors of Tracking (ST, GT/INT) and within-subject factors of Session (Day 1, Day 5; *Figure 1D*). STs show greater PCA score on Day 5 compared to GT/INTs (*Figure 1D*, PCA score: Session: $F_{(1,14)} = 67.3$, p<0.001, Session × Tracking: $F_{(1,14)} = 15.04$, p=0.002, Tracking: $F_{(1,14)} = 11.59$, p=0.004; post hoc, Day 5 ST vs. GT/INT presses: $t_{14} = .92$, p<0.001). Next, to confirm rats could discriminate the reinforced and non-reinforced lever cues, we examined the difference between CS + and CS− presses (Δ presses) and pokes (Δ pokes) using a mixed ANOVA with between-subject factors of Tracking (ST, GT/INT) and within-subject factors of Session (Day 1, Day 5, *Figure 1E*). ST rats show better discrimination for lever directed behavior (Δ presses) on Day 5 compared to GT/INTs (Session: $F_{(1,14)} = 35.75$, p<0.001, Session × Tracking: $F_{(1,14)} = 11.66$, p<0.001, Tracking: $F_{(1,14)} = 17.81$, p=0.001; post hoc, Day 5 ST vs. GT/INT presses: $t_{14}=3.93$, p=0.002). In contrast, GT/INTs show better discrimination for food cup-directed behavior (Δ pokes) on Day 5 compared to STs during the CS (*Figure 1F*: Session × Tracking: $F_{(1,14)} = 4.90$, p=0.044, Tracking: $F_{(1,14)} = 15.17$, p=0.002; post hoc, Day 5 ST vs. GT/INT pokes: $t_{14} = 3.92$, p=0.002).

To investigate the endogenous dBNST dopamine activity across PLA training, we used fiber photometry to monitor the fluorescent activity of the genetically encoded dopamine sensor, GRAB_DA (*Sun et al., 2018*). We see evidence of associative encoding during PLA (*Figure 2*). Both lever insertion and retraction/reward delivery increased dBNST GRAB_DA signals in ST and GT/INT rats (representative heat map and population average traces on Day 1 and Day 5 for STs in *Figure 2A* and GT/INTs in *Figure 2B*). To determine whether ST and GT/INT rats show differences in cue-evoked dopamine signals across acquisition of PLA, we compared the strength of CS+ onset (Δ lever extension area under curve (AUC) = (CS+) – (CS−) AUC; 2 s after CS onset) signals between Day 1 and Day 5 using a mixed ANOVA with between-subject factors of Tracking (ST, GT/INT) and within-subject factor of Session (Day 1, Day 5). While CS+ onset-evoked GRAB_DA signals increased across conditioning for both ST and GT/INT (*Figure 2C*, Session: $F_{(1,14)} = 19.69$, p=0.001) the magnitude of the CS+ signal increase differed between tracking groups (Session × Tracking: $F_{(1,14)} = 5.99$, p=0.028, Tracking: $F_{(1,14)} = 10.35$, p=0.006). Post hoc analyses revealed a greater cue-evoked dBNST GRAB_DA signal in ST compared to GT/INT on Day 5, which was not evident on Day 1 (Day 1: $t_{14} = 0.17$, p=0.87; Day 5: $t_{14} = 2.93$, p=0.011). Next, we asked whether GRAB_DA signals correlated with the tracking phenotype. We observed a positive correlation between Day 5 CS onset GRAB_DA signals and Day 5 PLA score (*Figure 2D*; $R^2=0.41$, p=0.009) but not Day 1 CS onset GRAB_DA signals and Day 1 PLA score (*Figure 2—figure supplement 1A*; $R^2=0.21$, p=0.09).

Next, we examined tracking differences in the sustained GRAB_DA signal between STs and GT/INTs throughout the duration of the CS, during which STs and GT/INTs show differences in lever and food cup-directed behaviors. We compared Day 1 vs. Day 5 CS+ maintained (Δ cue-period AUC = (CS+) – (CS−) AUC during the full 10 s CS lever insertion period) GRAB_DA signaling. CS+ maintained GRAB_DA signals increased across conditioning for both STs and GT/INTs (*Figure 2E*, Session: $F_{(1,13)} = 11.45$, p=0.005, Session × Tracking: $F_{(1,13)} = 3.07$, p=0.1, Tracking: $F_{(1,13)} = 16.5$, p=0.001). Like cue onset, we saw a strong positive correlation between Day 5 GRAB_DA signals during CS interaction and Day 5 PLA score (*Figure 2F*, $R^2=0.49$, p=0.004) but not Day 1 GRAB_DA signals and Day 1 PLA score (*Figure 2—figure supplement 1B*, $R^2=0.08$, p=0.3) suggesting that as rats display ST behavior, there's an increase in sustaineddBNST GRAB_DA signal.

Prior work shows that NAc dopamine shifts from US to CS after conditioning to a greater degree in STs compared to GTs (*Flagel et al., 2011*; *Lee et al., 2018*; *Saddoris, 2016*). Since we observed differences in CS evoked BNST GRAB_DA signals between STs and GT/INTs, we wanted to determine if there was similar tracking specificity in the US to CS shift for BNST GRAB_DA signals. We quantified the relative CS/US dynamics across conditioning using a difference score (Δ cue-reward AUC = (CS+) – (US) AUC for the 2 s after CS+ onset and reward delivery) and compared it between Day 1 and Day 5. We used a mixed ANOVA with between-subject factors of Tracking (ST, GT/INT) and within-subject factor of Session (Day 1, Day 5). The relative CS/US dynamics across PLA differed by tracking group (*Figure 2G*, Session: $F_{(1,14)} = 4.79$, p=0.046, Session × Tracking: $F_{(1,14)} = 8.9$, p=0.01). We found no tracking group differences in the (CS+) – (US) difference score on Day 1, but by Day 5 the CS/US difference score was greater in STs compared to GT/INTs (ST vs. GT/INT, Day 1: $t_{14}=−1.6$, p=0.13; ST vs. GT/INT, Day 5: $t_{14}=2.43$, p=0.029). While the correlation between (CS+) – (US) GRAB_DA signal

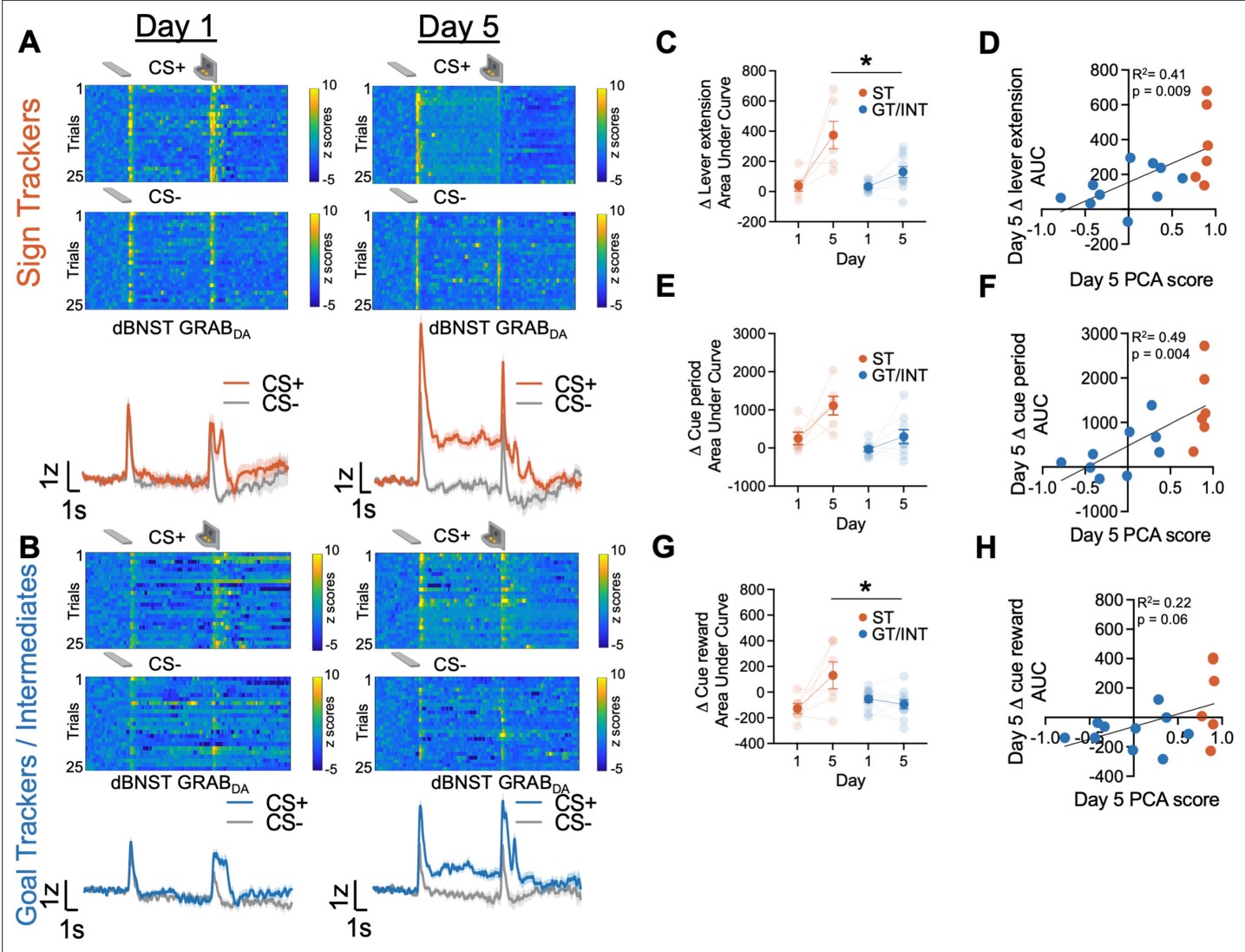

**Figure 2.** Dorsal bed nucleus of stria terminalis (dBNST) GRAB$_{DA}$ signals during Pavlovian lever autoshaping (PLA) between sign-trackers (STs) and goal-trackers/intermediates (GT/INTs). Representative heat maps illustrating GRAB$_{DA}$ signal changes (z-scores) during CS+ and CS− presentations on Day 1 (top left) and Day 5 (top right) and trial-averaged GRAB$_{DA}$ signal change (z-scored ΔF/F) during CS+ and CS− presentations on Day 1 (bottom left) and Day 5 (bottom right) in (**A**) STs and (**B**) GT/INTs. (**C**) Trial averaged quantification Δ lever extension ((CS+) – (CS−); 2 s) GRAB$_{DA}$ area under curve (AUC) between STs and GT/INTs. (**D**) Correlation between Day 5 Pavlovian conditioned approach (PCA) scores and Day 5 Δ lever extension AUC. (**E**) Trial averaged quantification of Δ cue period ((CS+) – (CS−); 10 s) in AUC during cue period between STs and GT/INTs. (**F**) Correlation between Day 5 PCA scores and Day 5 Δ cue period AUC. (**G**) Trial averaged quantification of Δ cue-reward ((CS+) – (US), 2 s) in AUC between STs and GT/INTs. (**H**) Correlation between Day 5 PCA scores and Day 5 change in Δ cue-reward AUC. Data are mean ± SEM. *p<0.05.

The online version of this article includes the following figure supplement(s) for figure 2:

**Figure supplement 1.** No correlation between Day 1 Pavlovian conditioned approach (PCA) scores and Day 1 bed nucleus of the stria terminalis (BNST) GRAB$_{DA}$ signals.

**Figure supplement 2.** Norepinephrine reuptake blocker, Reboxetine doesn't alter Pavlovian lever autoshaping (PLA) behavior or GRAB$_{DA}$ signals.

**Figure supplement 3.** Representative and population graph of signals along with behavior from rats that had correct fiber placement and viral expression but under 2z peak.

**Figure supplement 4.** Dopamine signals and behavior from rats that were removed from the study due to food cup entry artifact.

and Day 5 PCA scores was marginal (*Figure 2H*, $R^2$=0.22, p=0.06) there was no relationship between these measures on Day 1 (*Figure 2—figure supplement 1C*, $R^2$=0.025, p=0.56). Overall, these data indicate sign-tracking specific dBNST GRAB$_{DA}$ signals increase to Pavlovian cue onset and during cue-maintained sign-tracking behaviors, and back propagate from the reward to cue onset across conditioning.

## dBNST GRAB$_{DA}$ signals during PLA are specific to dopamine

Even though the GRAB$_{DA}$ construct we used is 15-fold more sensitive to dopamine than norepinephrine (NE); BNST NE plays an important role in motivated behaviors and dBNST receives dense noradrenergic input with relatively slow NE clearance measured in vivo (*Egli et al., 2005*; *Flavin and Winder, 2013*; *Park et al., 2009*; *Sun et al., 2020*). To validate that the signals we recorded during PLA were dopaminergic and not noradrenergic, we injected a NE reuptake inhibitor Reboxetine (1 mg/kg) 30 min prior to PLA. NE levels in the brain remain elevated at this dose for up to 3 hr peaking at ~20 min after injection (*Page and Lucki, 2002*). We found that Reboxetine injection did not alter behavior (*Figure 2—figure supplement 2B*) or increase BNST GRAB$_{DA}$ signal to lever extension or reward consumption (*Figure 2—figure supplement 2A, C and E*, Epoch: $F_{(1,12)}$ = 3.82, p=0.074, Epoch × Treatment: $F_{(1,12)}$ = 0.20, p=0.66, Treatment: $F_{(1,12)}$ = 0.21, p=0.66) compared to saline injection. Furthermore, there was no difference in the cue-interaction period between reboxetine and saline-injected conditions (*Figure 2—figure supplement 2D*; $t_6$=1.14, p=0.3). These data confirm that the signals we recorded during PLA are not sensitive to noradrenergic reuptake inhibition and are most likely due to fluctuations in DA signaling in the BNST.

## BNST dopamine encodes reward prediction error

After five Pavlovian autoshaping sessions, we conducted a Reward Prediction Error (RPE) session in which we randomly intermixed expected food reward trials with unexpected food reward delivery and omission trials. Expected reward (Expected) trials are identical to those delivered during training, with a 10 s CS+ lever insertion followed by retraction and food reward delivery. Unexpected reward (Positive) trials consist of randomly delivered food reward that is not signaled by a cue. Unexpected omission (Negative) trials consist of 10 s CS+ lever insertion and retraction, but no food reward is delivered. During these sessions, we monitored BNST GRAB$_{DA}$ signals to examine whether dopamine signals track errors in reward prediction (representative heat map for each trial type in *Figure 3A–C*; *Schultz et al., 1997*).

First, to determine whether BNST GRAB$_{DA}$ signals encode bidirectional reward prediction error (RPE), we compare signals on expected, positive and negative trials. Notably, because lever retraction occurs simultaneously with reward delivery, and sign- and goal-trackers may be in different locations at this time, we examine the signals during the 6 s (three 2 s bins) after reward delivery or omission, which captures the period corresponding to violations in reward expectations (*Figure 3D*). We performed a repeated measures ANOVA on z scores during the RPE session including Trial Type (Expected, Positive, Negative) and Bin (three 2 s bins (0–2 s, 2–4 s, 4–6 s)) as factors. We observed a difference in dBNST GRAB$_{DA}$ signaling between the three trial types in the bins following reward delivery/omission (*Figure 3D*, Bin: $F_{(2,72)}$ = 13.65, p<0.001, Bin × Trial Type: $F_{(4,72)}$ = 13.99, p<0.001, Trial Type: $F_{(2,36)}$ = 3.49, p=0.041). Post hocs confirm that in the second 2 s bin (i.e. 2–4 s) after reward delivery/omission, BNST GRAB$_{DA}$ signals differed from one another for all three trial types, Expected vs. Positive (population traces in *Figure 3E*; p=0.013), Expected vs. Negative (population traces in *Figure 3F*; p=0.043) and Positive vs. Negative (p=0.0004). Across all rats, we observe that dBNST GRAB$_{DA}$ signals reflect bidirectional reward prediction errors.

Then to determine whether there are tracking differences in dBNST RPE signals, we separately analyzed the z scores during RPE sessions in the two tracking groups. Again, we examine how GRAB$_{DA}$ signaling differs for the three trial types (expected, positive, negative) during the three 2 s bins after reward delivery (population traces for GT/INTs and STs *Figure 3G–H*). In GT/INT rats we observed main effects of Trial ($F_{(2,12)}$ = 8.2, p=0.006) and Bin ($F_{(2,12)}$ = 4.9, p=0.027) and a Trial × Bin interaction ($F_{(4,24)}$ = 25.7, p<0.001). GT/INT rats showed evidence for both positive RPE (Trial (Expected, Positive) × Bin interaction) ($F_{(2,12)}$ = 14.5, p=0.001) and negative RPE Trial (Expected, Negative) × Bin interaction ($F_{(2,12)}$ = 9.9, p=0.003; *Figure 3G* inset). In GT/INT rats, we next examined the time course and found dBNST GRAB$_{DA}$ signaling on both positive and negative trials differs from expected trials during the

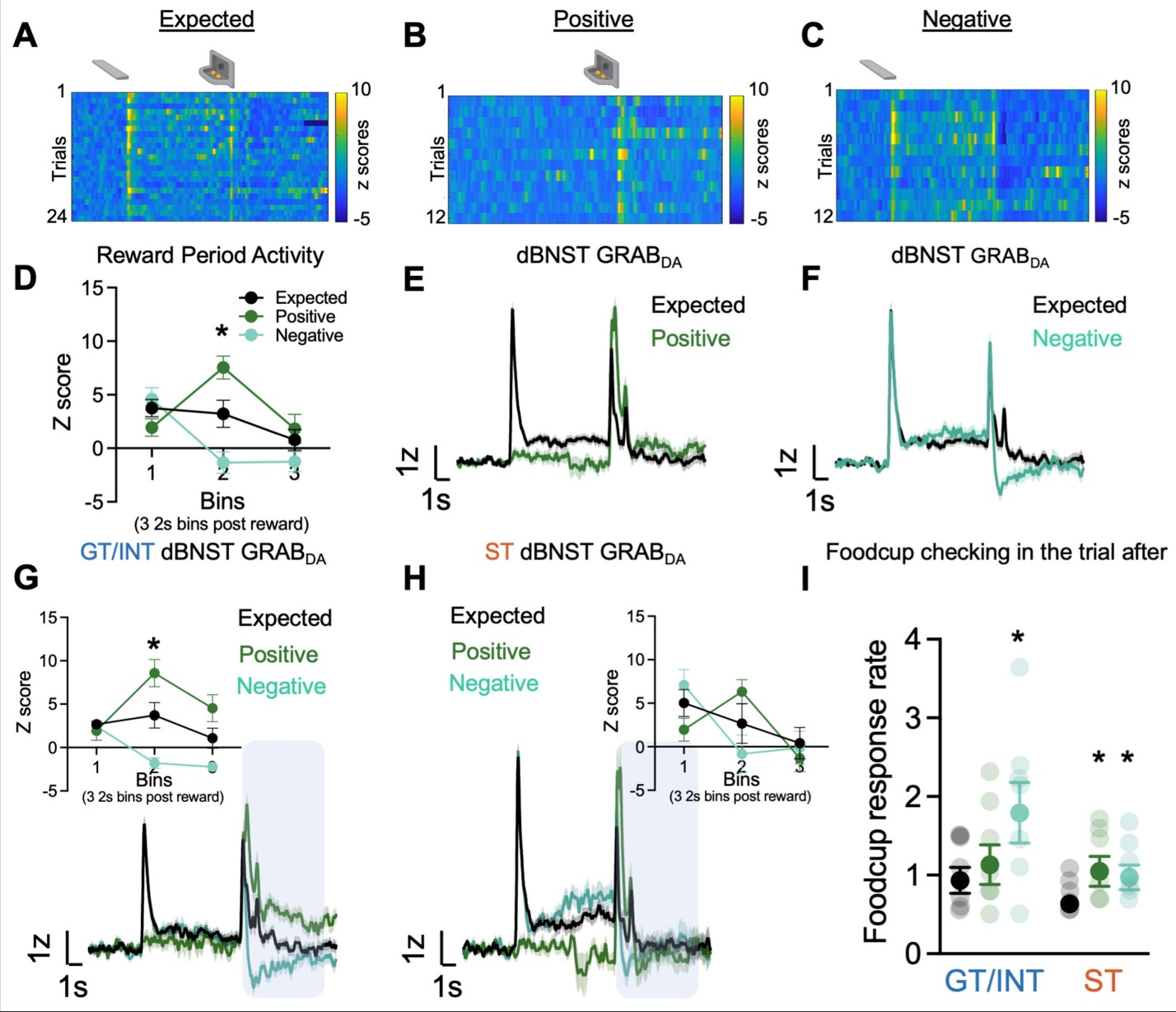

**Figure 3.** Individual differences in reward prediction error (RPE). (**A–C**) Representative heat maps during Expected, Positive (unexpected reward), and Negative (unexpected omission) reward trials. (**D**) Average binned z-scores (2 s bins) during Expected (N=13), Positive (N=13), and Negative (N=13) trials 6 s post reward delivery (bins 1–3). Trial-averaged GRAB$_{DA}$ signal change (z-scored ΔF/F) during (**E**) Expected vs. Positive trials and (**F**) Expected vs Negative trials in all rats. Trial-averaged GRAB$_{DA}$ signal change (z-scored ΔF/F) during all three trials and average binned z-scores (2 s bins) during Expected, Positive and Negative trials 6 s post reward delivery (bins 1–3) (inset) in (**G**) goal-trackers/intermediates (GT/INTs) (N=7) and (**H**) sign-trackers (STs) (N=6). (**I**) Average food cup checking response rate (responses/10 s) during 10 s pre-trial period on trial after expected, positive, and negative trials in GT/INTs vs. STs. Data are mean ± SEM, *p<0.05.

second 2 s bin (i.e. 2–4 s) after lever retraction/pellet delivery/omission (positive vs. expected p=0.04, negative vs. expected p=0.021). This suggests GT/INT rats show evidence for dBNST GRAB$_{DA}$ bidirectional RPE signaling.

In a parallel analysis in ST rats considering all trial types (Expected, Positive, Negative) we also observed the main effect of Bin ($F_{(2,10)}$ = 10.4, p=0.004) and Trial x Bin interaction ($F_{(4,20)}$ = 5.4, p=0.004). ST rats showed evidence for positive RPE (Trial (expected, positive) × Bin interaction $F_{(2,10)}$ = 6.8, p=0.014) but not negative RPE (Trial (expected, negative) x Bin interaction $F_{(2,10)}$ = 2.8, p=0.153, *Figure 3H* inset). Post hoc analyses in ST rats on the time-course failed to identify which bin GRABDA

signals distinguished by trial type, however, a planned analysis on the relevant second 2 s bin (i.e. 2–4 s after lever retraction/pellet delivery/omission) indicates a main effect ($F_{(2,10)}$ = 5.3, p=0.027) is marginally driven by Expected vs. Positive trial types ($F_{(1,5)}$ = 5.0, p=0.075 and not Expected vs. Negative trial types ($F_{(1,5)}$ = 2.0, p=0.216)). This analysis suggests ST rats fail to show evidence for dBNST GRAB$_{DA}$ bidirectional RPE signaling.

We collected behavioral data during RPE sessions and examined the pre-trial food cup checking rate (responses/10 s prior to CS onset/reward delivery) on the trial after a reward violation, during which prior studies establish invigoration of conditioned responses and orienting (*Holland and Gallagher, 1993a*, *Holland and Gallagher, 1993b*, *Calu et al., 2010*; *Roesch et al., 2010*). Rats increase their pre-trial food cup checking on trials after a reward violation (*Figure 3I*). We performed repeated measures ANOVA including factors of Trial Type (Expected, Positive, Negative) and Tracking (ST, GT/INT). While ST rats increase their pre-trial food cup checking after both positive (p=0.042) and negative (p=0.016) trials, GT/INTs only increase their pre-trial food cup checking following negative (p=0.013) trials (*Figure 3I*, Trial Type: $F_{(2,22)}$ = 10.9, p=0.001, Trial Type × Tracking: $F_{(2,22)}$ = 4.39, p=0.025, Tracking: $F_{(1,11)}$ = 1.77, p=0.21). These data indicate that STs and GT/INTs use different reward-seeking behavioral strategies following the violation of reward expectations.

## Reinforcer-specific but not general satiety attenuates cue-triggered GRAB$_{DA}$ signal

In the current and following sections, we report the number of ST and GT/INT rats for each experimental phase but do not report tracking differences due to decreased statistical power to detect group differences. Prior studies indicate that the midbrain and striatal dopamine system tracks motivational state through satiety-dependent changes in the magnitude of dopamine responses (*Cone et al., 2014*; *Hsu et al., 2018*; *Wilson et al., 1995*). Here, we determined whether the motivational state also decreases task-related BNST GRAB$_{DA}$ signals during lever autoshaping. After rats completed 25 trials of PLA along with the GRAB$_{DA}$ recordings, we sated rats (n=11, ST = 4, GT/INT = 7) on the training pellets presented in a ceramic ramekin in the homecage or presented a sham condition in which an empty ramekin was placed in the homecage for 30 min. Immediately after, we recorded GRAB$_{DA}$ signals during the remaining 25 trials of PLA sessions. First, we compared Δ presses and Δ pokes ((CS+) − (CS−)) between hungry and sated or hungry and sham conditions using two-way ANOVA with factors of State (Hungry, Sated) and Condition (Real, Sham). The number of presses differed based on the satiety condition compared to the hungry condition (State × Condition: $F_{(1,20)}$ = 9.65, p=0.006). Post hoc analysis revealed that rats sated on training pellets decreased lever presses predictive of food pellet reward (*Figure 4A* left, hungry vs sated presses: $t_{10}$=3.02, p=0.013; hungry vs. sham presses: $t_{10}$=−1.51, p=0.16). In contrast, the number of pokes generally but not differentially increased during the sated and sham conditions compared to the hungry condition (*Figure 4A* right, State: $F_{(1,20)}$ = 6.73, p=0.017, State × Condition: $F_{(1,20)}$ = 3.72, p=0.068). Similarly, we examined cue-evoked GRAB$_{DA}$ signal ((CS+) − (CS−); 2 s after cue onset) between hungry and sated or hungry and sham conditions using ANOVA with factors of State (Hungry, Sate) and Condition (Real, Sham). The differential change in lever presses was associated with the difference in cue-evoked GRAB$_{DA}$ signal during the sated and sham conditions compared to hungry condition (State × Condition: $F_{(1,20)}$ = 6.68, p=0.018). Post hoc analysis revealed that rats sated on pellets show a decrease in cue evoked GRAB$_{DA}$ signals but not in sham conditions (*Figure 4E* left, hungry vs. sated: $t_{10}$=2.71, p=0.022; hungry vs. sham: $t_{10}$=−0.95, p=0.35). While we observed a decrease in cue-triggered dopamine signals in sated conditions, there was no change in reward consumption-related dopamine signals in both sated and sham conditions (*Figure 4E* right, F's<0.52, p's>0.05). These results further bolster our finding that BNST GRAB$_{DA}$signals encode cue-outcome associations, which, similar to striatal dopamine signaling, is blunted when the animal has reduced motivational drive (*Cone et al., 2014*; *Wilson et al., 1995*).

Next, we examined whether the reduction in cue evoked GRAB$_{DA}$ signal is specific to the training pellet or whether it is sensitive to a general satiety state by sating rats on homecage chow (n=7, ST = 3, GT/INT = 4). We conducted analyses similar to the pellet satiety experiment. When we sated rats on chow, the number of presses differed based on the satiety condition compared to the hungry condition (State × Condition: $F_{(1,12)}$ = 5.86, p=0.032), however, there was no change in cue-evoked GRAB$_{DA}$ signals (*Figure 4B, D, F* F's<1.8, p's>0.05). Similarly, the number of pokes also differed based on the satiety condition compared to the hungry condition (State × Condition: $F_{(1,12)}$ = 9.61, p=0.009).

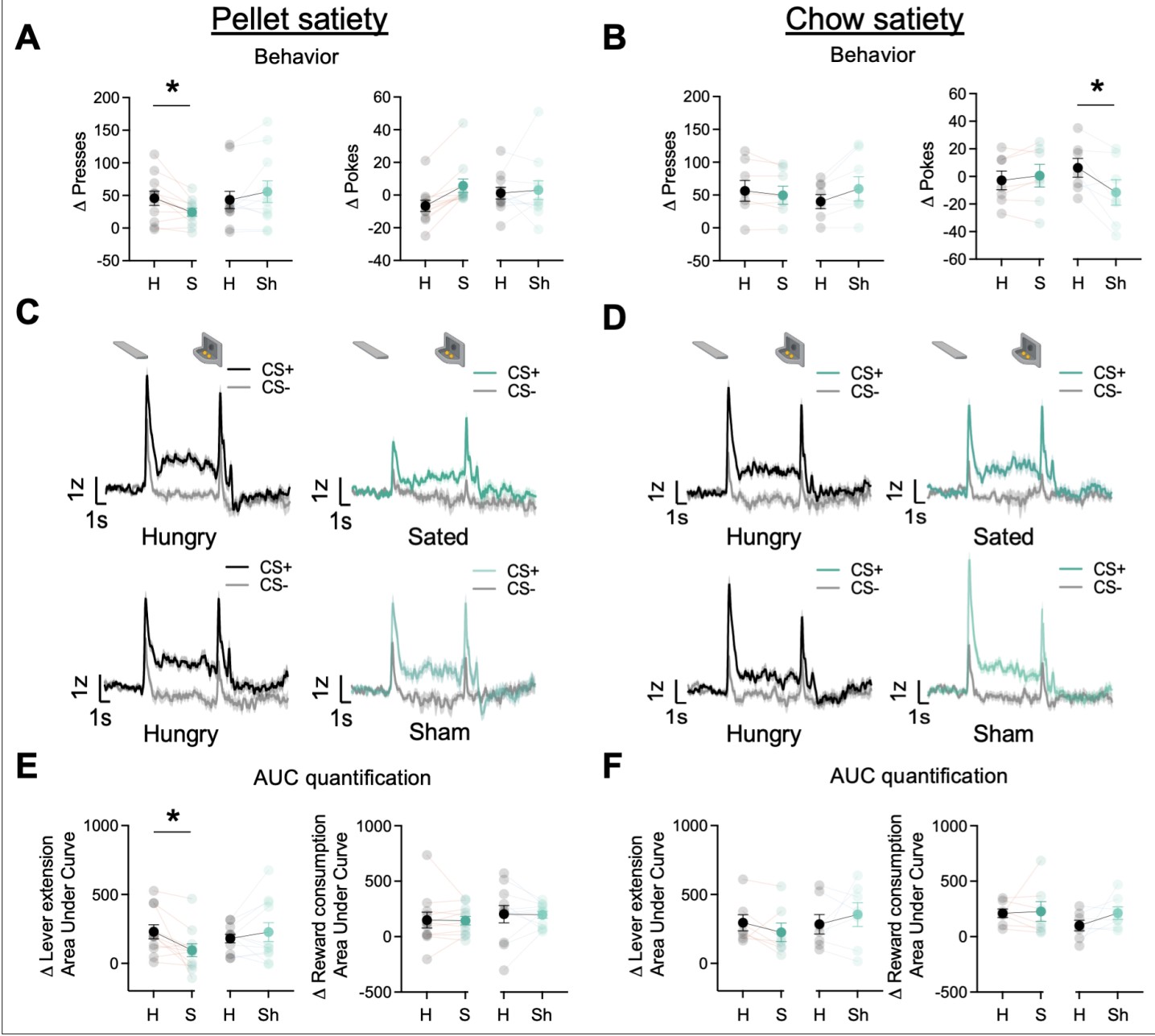

**Figure 4.** Dorsal bed nucleus of the stria terminalis (dBNST) GRAB_DA signals attenuate after reinforcer-specific but not general satiety. (**A**) Average Δ Presses (CS+) – (CS−) (left) and average Δ pokes (CS+) – (CS−) (right) when rats were either sated on training food pellets in the ramekin or sham-sated (ramekin only). (**B**) Average Δ Presses (CS+) – (CS−) (left) and average Δ pokes (CS+) – (CS−) (right) when rats were either sated or sham-sated on homecage chow. (**C**) Trial-averaged GRAB_DA signal change (z-scored ΔF/F) during CS+ and CS− presentations when rats were hungry versus sated (top) and when rats were hungry versus sham-sated (bottom) on food pellets and (**D**) on homecage chow. (**E**) Trial average quantification of change (CS+) – (CS−) in an area under GRAB_DA z-scored curve (AUC) during lever extension (2 s) (left) and reward consumption (right) between food pellet sated and sham and (**F**) between homecage chow sated and sham conditions. Data are mean ± SEM, *p<0.05. H=Hungry, S=Sated, Sh = Sham conditions.

Post hoc analysis revealed that rats decreased their poking for sham compared to hungry ($t_6$=2.87, p=0.03). This is presumably due to a concurrent non-significant increase in lever presses (sham sate presses: $t_6$=−1.92, p=0.1). But this decrease in food cup pokes was not accompanied by a change in reward consumption evoked GRAB_DA signal (F's<1.3, p's>0.05). These results suggest that when rats are sated on the outcome associated with the Pavlovian cue, there is an attenuation in GRAB_DA signals while a general satiety doesn't attenuate cue responding or GRAB_DA signals.

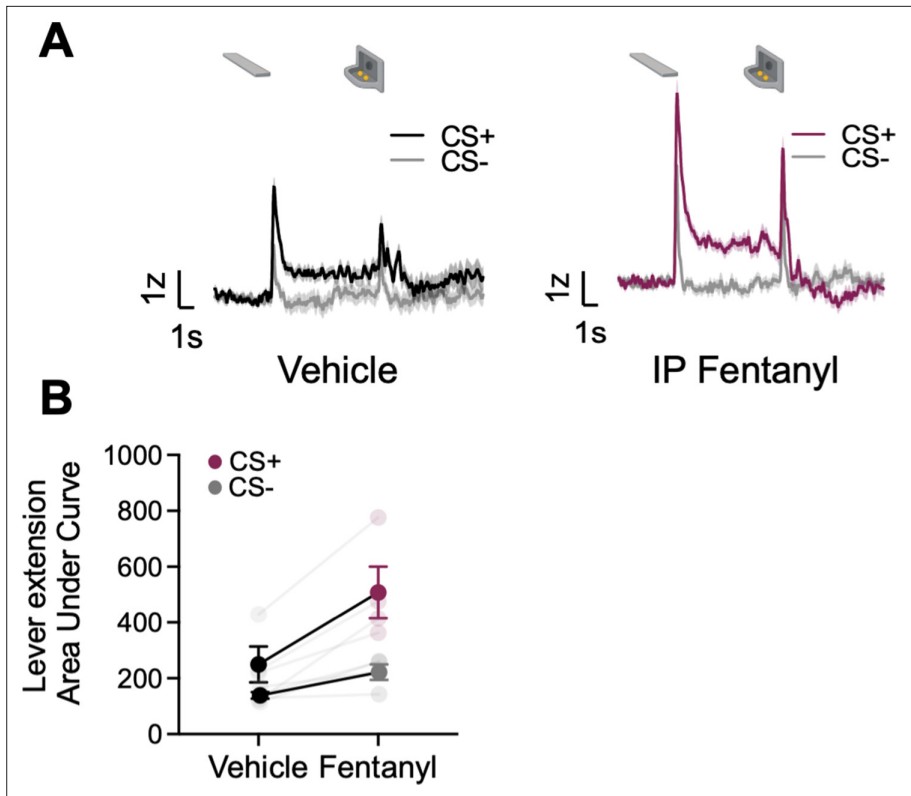

**Figure 5.** Systemic administration of fentanyl results in the potentiation of dorsal bed nucleus of the stria terminalis (dBNST) dopamine. (**A**) Trial-averaged GRAB$_{DA}$ signal change (z-scored ΔF/F) when rats were injected with vehicle (left) or fentanyl (right) during Pavlovian lever autoshaping (PLA) (**B**) Trial average quantification of the area under GRAB$_{DA}$ z-scored curve (AUC) during CS+ and CS− lever extension (2 s) between vehicle and fentanyl conditions. Data are mean ± SEM, *p<0.05.

## Systemic fentanyl administration boosts GRAB$_{DA}$ signals to reward-related cues

Opioids potentiate NAc activity and NAc DA responses to natural rewards and natural reward-associated cues (*Bassareo et al., 2013*; *Castro and Berridge, 2014*; *Mahler et al., 2007*; *Peciña and Berridge, 2005*). Here, we sought to determine whether opioids also potentiate task-related BNST GRAB$_{DA}$ signals during natural reward seeking during lever autoshaping. We recorded GRAB$_{DA}$ signals in a subset of rats (n=4, ST=2, GT/INT=2) during PLA after i.p injection of synthetic μ-opioid agonist, fentanyl, 5 μg/kg (population average traces for saline and ip fentanyl injection in *Figure 5A*). We observed the main effects of Treatment (vehicle, fentanyl) and CS (CS+, CS−), but the interaction was not significant, indicating that cue discrimination is maintained with systemic fentanyl injections, which generally potentiate DA signaling in the dBNST (*Figure 5B*, CS: $F_{(1,6)}$ = 24.42, p=0.003, Treatment: $F_{(1,6)}$ = 7.16, p=0.037, CS × Treatment: $F_{(1,6)}$ = 2.8, p=0.15).

## Sex as a biological variable

We use both male and female rats and have analyzed our photometry data from Pavlovian autoshaping, RPE, satiety, and fentanyl test sessions using Sex instead of Tracking as a factor. We observed no main effects of Sex or interaction between Sex and any other factor.

## Discussion

Using a fluorescent dopamine sensor, GRAB$_{DA}$, we characterized phasic dBNST dopamine signals during a range of appetitive Pavlovian conditions including lever autoshaping, reward violations, specific satiety, and fentanyl injections during PLA. We found that dBNST dopamine signals are

enhanced in STs compared to GT/INTs during cue presentation and shift from reward to cue across conditioning in STs but not in GT/INTs. Furthermore, dBNST dopamine signals encode bidirectional reward prediction error and are greater in GT/INTs than in STs following reward violations. Additionally, dBNST dopamine signals decrease to cues when rats are sated on food pellets associated with the cue but not when sated on homecage chow. Systemic fentanyl injections do not disrupt dBNST cue discrimination but generally potentiate dBNST dopamine signals.

Pharmacological studies establish that dopamine signaling in the dBNST maintains responding to sucrose and ethanol rewards and regulates the reinforcing properties of cocaine (*Eiler et al., 2003*; *Epping-Jordan et al., 1998*). Microdialysis and voltammetry studies show that natural and drug rewards, including opioids, increase DA in the BNST (*Carboni et al., 2000*; *Park et al., 2012*; *Park et al., 2013*). Although dBNST dopamine is important for a variety of appetitive motivated behaviors, little is known about cue-evoked dopamine signaling and its role in cue-triggered motivation. A recent study showed BNST GRAB$_{DA}$ signals are associated with both cues and rewards (*Lin et al., 2020*). Our data extend these findings by showing individual differences in CS- and US-evoked BNST dopamine signaling during Pavlovian conditioning. We also demonstrate that CS-evoked BNST DA signals are state-dependent and outcome-specific.

Consistent with prior studies, we observed individual differences in sign- and goal-tracking behaviors elicited by the CS (*Boakes, 1977*; *Hearst and Jenkins, 1974*; *Nasser et al., 2015*; *Robinson et al., 2014*). Accompanied by this behavioral variation, we observed tracking differences in GRAB$_{DA}$ signals to CS onset and differences in dopamine signal transfer from US to CS, both of which were stronger in sign-tracking compared to goal-tracking and intermediate rats. We observed a relationship between CS-maintained GRAB$_{DA}$ signal and PCA scores, indicating sign-tracking approach and interaction with the lever cue is associated with heightened dBNST GRAB$_{DA}$ signaling. These findings for the dBNST dopamine signal are consistent with prior tracking differences in NAc dopamine signals during Pavlovian lever autoshaping (*Flagel et al., 2011*). We also find that only in ST rats did GRAB$_{DA}$ signals adhere to *Sutton and Barto, 2018* reinforcement learning algorithm, which states that after learning, reward-evoked signals are temporally transferred back to antecedent cues predicting reward delivery (*Nasser et al., 2017*; *Sutton and Barto, 2018*). Consistent with this, we observed an increase in sustained GRAB$_{DA}$ signal during the entire 10 s CS interaction period on Day 5 of PLA training compared to Day 1. Sustained BNST GRAB$_{DA}$ signals during the cue interaction period could reflect a number of processes, including (1) ongoing lever interaction, (2) the incentive value gain of the CS, (3) the strength of CS-US association, and/or (4) the back-propagating US to CS signal. Our results suggest that dopamine signaling differences between STs and GTs are not just limited to NAc and could be present across a distributed network receiving dopaminergic projections.

To adapt to environmental changes and learn about future rewards, dopaminergic neurons calculate reward prediction errors (RPE) (*Nasser et al., 2017*; *Schultz et al., 1997*; *Watabe-Uchida et al., 2017*). Here, we examined if BNST GRAB$_{DA}$ signals encode RPE and whether there are individual differences in dBNST GRAB$_{DA}$ signals and behavioral strategies following violations of reward expectations. We found that dBNST GRAB$_{DA}$ signals follow the classical bidirectional prediction error signal such that the signals increased following unexpected reward delivery and decreased following unexpected reward omission. Consistent with attention to learning theories and empirical studies, we observed that rats increase their food cup checking behavior on a trial after a positive or negative reward violation (*Calu et al., 2010*; *Pearce and Hall, 1980*; *Roesch et al., 2010*). Sign-tracking rats increase food cup checking on trials after both unexpected reward delivery and omission, whereas GT/INTs increase food cup checking only after reward omission. Behaviorally this suggests GT/INT rats may be more sensitive to negative reward violations than positive, which is consistent with their sensitivity to outcome devaluation and their insensitivity to conditioned reinforcement (*Keefer et al., 2020*; *Keefer et al., 2022*; *Kochli et al., 2020*; *Morrison et al., 2015*; *Nasser et al., 2015*; *Robinson and Flagel, 2009*; *Smedley and Smith, 2018*). Such excitatory behavioral responses (more checking for both increases and decreases in reward) before the trial are evidence for an incremental attentional processes, which reflect enhanced attention to environmental predictors for the purpose of increasing the rate of learning for either excitatory or inhibitory associations (*Calu et al., 2010*; *Holland and Gallagher, 1993a*; *Pearce and Hall, 1980*; *Roesch et al., 2007*; *Roesch et al., 2010*). Notably, reward prediction errors are critical for such enhanced attentional processes, and the theoretical instantiation of incremental attention for learning about positive and negative reward violations takes the absolute

value of RPE signals into account (*Pearce and Hall, 1980*). Prior work establishes the involvement of other amygdala nuclei, namely the basolateral and central nuclei of the amygdala for encoding unidirectional prediction error signals that track enhanced attention after reward violations (*Calu et al., 2010*; *Roesch et al., 2010*). Midbrain dopamine signaling is required for such attentional encoding in the basolateral amygdala (*Esber et al., 2012*). Here, we identify bidirectional dopamine encoding of positive and negative reward violations in an extended amygdala nuclei, the dBNST. GT/INTs showed evidence for bidirectional RPE in the dBNST DA signal, which may be important for enhancing attention signals in downstream areas.

BNST receives heavy dopaminergic afferents from the A10 Ventral Tegmental Area (VTA) and A10dc ventral periaqueductal gray/dorsal raphe (vPAG/DR) dopaminergic cell groups, and to a lesser extent from the substantia nigra pars compacta and the retrorubral nucleus (*Daniel and Rainnie, 2016*; *Hasue and Shammah-Lagnado, 2002*; *Melchior et al., 2021*; *Meloni et al., 2006*; *Vranjkovic et al., 2017*). While VTA and SNc dopamine neurons classically encode bidirectional reward prediction error signals, vPAG dopamine and its projections unidirectionally encode rewarding and aversive outcomes, suggesting salient event detection (*Berg et al., 2014*; *Lin et al., 2020*; *Nasser et al., 2017*; *Schultz et al., 1997*; *Walker et al., 2020*; *Watabe-Uchida et al., 2017*). Different aspects of the dBNST DA signaling we observed lead us to postulate both dopamine projections may be contributing. For the bidirectional RPE, we observed in dBNST, we predict that VTA dopaminergic projections are the source of dopamine during reward violations. In contrast, for the greater CS signaling in ST compared to GT/INT rats may reflect salient features of the CS that support the attracting and reinforcing properties of cues in sign-tracking rats, which may also be supported by vPAG/DR→BNST dopamine. Future studies are needed to identify the extent to which VTA and PAG/DR dopaminergic inputs contribute to the BNST DA signals observed here. Dissecting the role of each dopaminergic input in driving cue and reward-related signaling and behavior will inform whether these circuits work synergistically or in competition to influence appetitive behaviors (*Lin et al., 2020*; *Park et al., 2012*; *Park et al., 2013*). Anatomical studies indicate that a substantial proportion of putative dopaminergic projections to BNST originate in the vPAG/DR (*Hasue and Shammah-Lagnado, 2002*; *Meloni et al., 2006*). Prior work has established glutamate and dopamine co-release from the vPAG/DR projection to BNST, positioning this input to directly influence BNST synaptic plasticity and associated behaviors (*Li et al., 2016*). Regardless, the dopamine dynamics reported here for BNST resemble those previously reported for nucleus accumbens in related behaviors (*Clark et al., 2013*; *Flagel et al., 2011*; *Hart et al., 2014*; *Saddoris et al., 2015*; *Saddoris et al., 2016*), suggesting a potential role for VTA DA in shaping BNST DA signaling. Notably, NAc DA also shows greater CS-evoked, and a greater shift from US to CS- evoked DA in ST compared to GT (*Flagel et al., 2011*). To our knowledge, tracking-related differences in bidirectional RPE signaling in the NAc have not been systematically tested, however, the bidirectional error encoding we observe across all rats is consistent with prior NAc voltammetry studies (*Hart et al., 2014*). Here, we report that GT/INT, but not ST, show evidence for bidirectional RPE DA signaling in the BNST. Whether this is also the case for NAc DA signaling remains an open question. Consistent with our findings, short inter-trial-intervals (ITI, similar to what we employ here) during autoshaping promote both classic NAc DA RPE signaling and goal-tracking, whereas longer ITIs promote NAc DA CS-salience signaling and sign-tracking (*Lee et al., 2018*). Pharmacology studies show D1 receptors and NAc DA signaling drive CS-salience in sign-trackers (*Chow et al., 2016*; *Saunders and Robinson, 2012*). The potentiating effects of hunger and systemic fentanyl injections on BNST DA signals observed here are in line with effects observed for NAc DA (*Bassareo et al., 2013*; *Castro and Berridge, 2014*; *Cone et al., 2014*; *Mahler et al., 2007*; *Peciña and Berridge, 2005*; *Wilson et al., 1995*). Notably, NAc primarily receives input from the VTA, whereas the BNST receives DA inputs from VTA and vPAG/DR. The tracking-specific differences in BNST dopamine signaling during simple appetitive approach and reward violations observed here suggest either (1) distinct contributions of VTA and vPAG/DR to dopamine signaling observed in BNST and/or (2) individual differences in the engagement of DA systems that bias towards CS-salience or RPE processes (*Chow et al., 2016*; *Lee et al., 2018*). Consideration of tracking-specific dopamine signaling differences in future studies that employ projection-specific manipulations will aid in interpreting each projection's contribution to BNST dopamine signaling and behavior.

A methodological limitation of the current approach is that variations in the expression of fluorescent sensor and/or fiber placement along a gradient of DA input to BNST could potentially influence

the magnitude of GRAB$_{DA}$ measurements. Our fiber placements were largely consistent (~73% at the level of bregma) and overlapped with the densest area of viral expression of the fluorescent sensor. At this level of BNST where we measured the majority of GRAB$_{DA}$ signals, there is heavy vPAG/DR DA input and to a lesser extent VTA input (*Hasue and Shammah-Lagnado, 2002*). Other anatomical and/or functional studies that target BNST up to 0.2 mm anterior or posterior to bregma also observe substantial putative dopaminergic input from vPAG/DR (*Meloni et al., 2006*; *Yu et al., 2021a*). Regardless, some rats presented in *Figure 2—figure supplement 3* were excluded that had sufficient viral expression and fiber placement, but that did not show evidence of dopamine binding in the dBNST during these Pavlovian tasks. While we are limited from drawing conclusions from negative data, such individual differences in extended amygdala dopamine signaling may be important for interpreting differences in appetitive behaviors. In addition, we analyzed signals that were significantly different from the baseline (greater than 2z scores) in our behavioral window. We might have missed some behaviorally relevant signals due to this restriction. Future studies with control GRAB$_{DA}$ virus are needed to determine how large a signal can be expected from artefactual sources (blood flow, autofluorescence, movement, etc).

The BNST is a sexually dimorphic brain region (*Hisasue et al., 2010*; *Shah et al., 2004*; *Tsuneoka et al., 2017*), highlighting the necessity of studying both sexes to fully understand the contribution of BNST DA to motivated behavior. Dopaminergic projections from vPAG/DR→BNST play sex-specific roles, with pathway activation associated with distinct pain and locomotor behavioral changes for males and females, respectively (*Yu et al., 2021b*). We used both male and female rats in the present study and analyzed our BNST DA photometry data from Pavlovian autoshaping, RPE, and satiety test sessions using Sex instead of Tracking as a factor. While we observed no sex effects here, prior studies establish BNST-mediated sex differences in pain and locomotor behaviors as well as in opioid withdrawal (*Luster et al., 2020*; *Yu et al., 2021a*; *Yu et al., 2021b*). While there is limited evidence for sex differences in the incubation of fentanyl seeking (a form of relapse), we find this effect to be dependent on dBNST CRFR1 receptor signaling (*Gyawali et al., 2020*; *Reiner et al., 2019*; *Reiner et al., 2020*). Drug-induced synaptic plasticity in the dBNST requires both dopamine and CRF and molecular and electrophysiology studies suggest that DA increases CRF release in the dBNST (*Day et al., 2002*; *Kash et al., 2008*). Given the known role of sex differences in CRF-induced relapse and opioid withdrawal, it is critical to include both sexes when studying BNST DA and CRF systems (*Buffalari et al., 2012*; *Luster et al., 2020*).

To our surprise, we found evidence for outcome-specific state-dependent BNST GRAB$_{DA}$ signaling. Consistent with our prior studies, we found that rats decreased their lever responding only when they were sated on food pellets specifically associated with the lever cue, but not when sated on homecage chow (*Keefer et al., 2020*; *Kochli et al., 2020*). Similarly, we observed decreased cue-evoked BNST GRAB$_{DA}$ when rats were sated on food pellets but not when they were sated on chow. All rats ate all their pellets during these reinforced sessions, and we did not see any change in GRAB$_{DA}$ signals during reward consumption when sated on either food pellets or chow. Prior studies report a similar decrease in cue-evoked dopamine signals in the basolateral amygdala and dopaminergic neuron activity in the dorsal raphe during satiety (*Cho et al., 2021*; *Lutas et al., 2019*). Based on these studies that manipulated state using hunger or satiety, we expected dopamine signals to generally decrease to cues both when sated on chow or training pellets, but we found BNST dopamine signals only decreased when sated on the training pellet associated with the cue. However, other studies find evidence for sensory-specific signaling in dopamine function and signaling (*Sharpe et al., 2017*; *Takahashi et al., 2017*). This suggests BNST DA signals may carry sensory-specific information that is critical for higher-order learning processes (*Burke et al., 2007*; *Burke et al., 2008 Keefer et al., 2021*; *Lichtenberg et al., 2017*; *Lichtenberg et al., 2021*; *Malvaez et al., 2015*; *Malvaez et al., 2019*; *Sias et al., 2021*; *Sharpe et al., 2017*; *Takahashi et al., 2017*).

Studies show elevated BNST dopamine, dopamine-induced plasticity, and dopamine-mediated seeking behavior during and after drug administration (*Carboni et al., 2000*; *Eiler et al., 2003*; *Epping-Jordan et al., 1998*; *Kash et al., 2008*; *Krawczyk et al., 2013*; *Krawczyk et al., 2011a*; *Krawczyk et al., 2011b*; *Melchior et al., 2021*; *Stamatakis et al., 2014*). We extend these findings by reporting that systemic fentanyl injections do not disrupt dBNST cue discrimination but generally potentiate dBNST dopamine signals. The present study supports the need for future work aimed at

fully characterizing drug-induced changes to dBNST DA cue and reward encoding during natural and opioid reward seeking.

Dopamine projections to the BNST are concentrated in the dBNST and synapse specifically onto the CRFergic neurons (*Meloni et al., 2006*; *Phelix et al., 1994*). Molecular and electrophysiology studies suggest that dopamine increases local CRF release in the dBNST and drug-induced synaptic plasticity in the dBNST requires both dopamine and CRF (*Day et al., 2002*; *Kash et al., 2008*; *Silberman et al., 2013a*). These anatomical and ex vivo physiology studies suggest dopamine and CRF are critically interacting to drive reward and stress-related behaviors. Indeed, our prior work indicates that CRF receptor activation in the dBNST is necessary for CS-triggered opioid relapse (*Gyawali et al., 2020*). Furthermore, dBNST dopamine receptor activation decreases blood corticosterone levels in mice suggesting that an increased dopamine response in the dBNST could serve as an anxiolytic signal, which could promote continued drug seeking (*Daniel and Rainnie, 2016*; *Kash et al., 2008*; *Melchior et al., 2021*; *Meloni et al., 2006*).

The present findings add substantially to the role of dBNST dopamine in motivated behaviors, providing a comprehensive characterization of endogenous dBNST dopamine dynamics in cue-induced behaviors under several natural and drug reward conditions. The fluorescent dopamine sensor GRAB_DA is a useful tool for studying real-time BNST DA dynamics in the context of motivated behaviors (*Lin et al., 2020*; *Sun et al., 2020*).

# Materials and methods

### Key resources table

| Reagent type (species) or resource | Designation | Source or reference | Identifiers | Additional information |
|---|---|---|---|---|
| Transfected construct (*H. sapiens*) | AAV9.hsyn.DA4.4.eyfp | WZ Biosciences | h-D03 | Titer >1.0 × 10E13GC/mL |
| Chemical compound, drug | Fentanyl | Cayman Chemicals | Cat: 22659 | |
| Chemical compound, drug | Reboxetine Mesylate | MedChemExpress | Cat: HY-14560C | |
| Chemical compound, drug | TCS | Access Technologies | Cat: TCS-04 | |
| Chemical compound, drug | Paraformaldehyde | Sigma | Cat: P6148 | |
| Software, algorithm | MED-PC IV | Med Associates | RRID: SCR_012156 | Version: IV |
| Software, algorithm | Excel | Microsoft | RRID: SCR_016137 | |
| Software, algorithm | SPSS | IBM | RRID: SCR_019096 | Version: 26 |
| Software, algorithm | Matlab | Mathworks | RRID: SCR_001622 | Version: 2020 a |
| Software, algorithm | Graphpad Prism | Graphpad Software | RRID: SCR_002798 | Version: 9 |
| Software, algorithm | Synapse Software | Tucker-Davis Technologies | RRID: SCR_006495 | |
| Other | LED Driver | ThorLabs | Cat: DC4100 | LED Driver capable of driving high-power four-wavelength LED sources simultaneously with a current range between 0 and 1000 mA |
| Other | Fluorescence Minicube | Doric | Cat: ilFMC4-G2_IE(400--410)_E(460-490)_F(500--550)_S | Fluorescence Mini Cube with 4 ports: one port for the functional fluorescence excitation light, one for the isosbestic excitation, one for the fluorescence detection, and one for the sample |
| Other | Fiber optic patchcord | Doric | D202-4094-3 | MFP_400/430/LWMJ-0.48_3 m_FCM-MF2.5 |
| Other | Fiber optical cannula | ThorLabs | Cat: CFMC54L10 | Ceramic ferrule Ø400 μm core, 0.50 NA fiber that is flat cleaved to a length 10 mm |
| Other | Metabond powder | Parkell | Cat: S396 | See *Virus and fiber optic implantation surgery* for more details |

*Continued on next page*

*Continued*

| Reagent type (species) or resource | Designation | Source or reference | Identifiers | Additional information |
|---|---|---|---|---|
| Other | Metabond quick base | Parkell | Cat: S398 | See *Virus and fiber optic implantation surgery* for more details |
| Other | Metabond catalyst | Parkell | Cat: S371 | See *Virus and fiber optic implantation surgery* for more details |
| Other | Dental Cement | DenMat | Cat: 034524101 | See *Virus and fiber optic implantation surgery* for more details |
| Other | Dental Cement Catalyst | DenMat | Cat: 4506 | See *Virus and fiber optic implantation surgery* for more details |
| Other | Sucrose pellets | Test Diet | 5TUL; Cat: 1811155 | Purified ingredient rodent tablet, protein: 20.6%, fat: 12.7%, carbohydrate: 66.7% |

## Subjects

We used 8-weeks-old male and female Sprague Dawley rats (Charles River, n=42) weighing >250 g before surgery. After surgery, we individually housed the rats and maintained them under a reversed 12:12 hr light/dark cycle (lights off at 9 AM). We estimated the sample size based on prior studies (*Bacharach et al., 2018*; *Kochli et al., 2020*) and pilot experiments. Each primary experiment was replicated in at least one additional cohort. Investigators were blinded to the tracking phenotype until the end of the experiments. We performed all experiments in accordance with the 'Guide for the care and use of laboratory animals' (8th edition, 2011, US National Research Council) and the University of Maryland Institutional Animal Care and Use Committee approved all experimental procedures. We excluded rats because of a lack of viral expression (n=4), incorrect fiber optic placements (n=6), and headcap loss (n=4). Additionally, rats (n=4) presented in *Figure 2—figure supplement 3* were excluded that had sufficient viral expression and fiber placement but did not show robust photometry signals in the dBNST during CS+ presentation by day five of PLA training (see *Photometry Analysis* subsection for more details). Finally, we excluded rats (n=8) presented in *Figure 2—figure supplement 4* that showed food cup entry artifacts before we optimized our photometry setup. The artifacts resulted in the loss of signal due to the patch cord hitting the wall of the food cup.

## Virus and fiber optic implantation surgery

We anesthetized 9-week-old rats with isoflurane (4.5% induction, 2–3% maintenance) and placed them in a stereotaxic frame. We maintained stable body temperature with a heating pad and administered pre-operative analgesic carprofen (5 mg/kg, s.c) and lidocaine (10 mg/mL at the site of incision). We made a scalp incision and drilled a hole above left dBNST AP = 0.0 or –0.1 from bregma, ML = +3.5, DV = –6.75 or –6.8 at 16° from midline for viral injection, and DV = –6.6 or –6.7mm relative to the skull for fiber implantation. In addition, we also drilled three holes anterior and posterior to attach anchor screws. We lowered the 5 μL Hamilton syringe unilaterally into the dBNST and injected AAV9. hsyn.DA4.4.eyfp (1.14 × 10^{14} GC/mL; WZ Biosciences) via a micropump at a volume of 0.7–1 μL over 10 min. We implanted the fiber optic (ThorLabs CFMC54L10, 400 μm, 0.50 NA, 10 mm) 0.1 mm or 0.15 mm above the virus injection site. We anchored the fiber optic to the skull using dental cement (Metabond and Denmat) and jeweler screws. We handled the rats at least three times a week after surgery before starting behavioral and photometry sessions.

## Apparatus

We conducted behavioral experiments in operant chambers housed in sound-attenuating cabinets (Med Associates). Each chamber had one white house light that was illuminated during the entire session. On the opposite wall, two retractable levers (CS+ and CS−, right or left location counterbalanced) were located on either side of the food cup. The food cup was attached to a programmed

pellet dispenser that delivered 45 mg training pellets (Testdiet, 5 TUL, protein 20.6%, fat 12.7%, carbohydrate 66.7%).

## Pavlovian lever autoshaping (PLA)

We conducted all training sessions during the dark phase. Schematic of our behavioral design can be found in *Figure 1A*. Five weeks after viral injection surgery, we maintained rats at 90% of ad libitum body weight during all behavioral sessions unless noted otherwise. Prior to the PLA training, we exposed rats to 25 magazine training trials divided into three sessions to acclimatize rats to the operant box and fiber optic cables. The three sessions consisted of 7, 8, and 10 trials, respectively in which two food pellets (US) were delivered, 0.5 s apart using a variable interval (VI) the 60 s (50–70 s) schedule. After magazine training sessions, we trained rats in five 46 min PLA sessions. Each session consisted of 25 reinforced (CS+) and 25 non-reinforced (CS−) lever presentation trials on a mean VI 45 s (35–55 s) schedule (*Figure 1B*). Each CS+ trial consisted of the insertion and retraction of a lever for 10 s followed by delivery of two food pellets, 0.5 s apart. CS− trials consisted of insertion/retraction of another lever, but no US delivery. We recorded the food cup and lever approach during the 10 s CS interaction and calculated a Pavlovian Conditioned Approach (PCA) score (*Berg et al., 2014*; *Meyer et al., 2012*). We use the PCA score as a comprehensive measure of individual differences in PLA that accounts for contact, latency, and probability differences. We used each rat's Days 4 and 5 average PCA score to determine whether they are sign-trackers (avg PCA score ≥0.5, ST) or goal-trackers/intermediates (avg PCA score <0.5, GT/INT).

## Reward prediction error (RPE) probe sessions

After five PLA sessions, we gave rats (n=13) one session in which we violated rats' reward expectations to probe for reward prediction error signaling. During this session, only the CS+ lever was presented, and rats received 48 trials divided into three different trial types presented in pseudorandom order. In the 'expected reward' condition, we gave 24 reinforced CS+ → US trials (50% of total trials). In the 'unexpected reward or positive' condition, we delivered two food pellets (US) randomly during the intertrial interval period without the predictive CS+ (12 trials, 25% of total trials). Finally, in the 'unexpected reward omission or negative' condition, we delivered the CS+, but omitted the US (12 trials, 25% of total trials) (*Patriarchi et al., 2018*).

### Satiety test

After the RPE session, we trained a subset of rats (n=11) in PLA for two more days when rats were either sated on food pellets or hungry. On the first day, we gave half the rats 30 g of the training food pellets in a ramekin for 30 min (pellet-sated condition) in their home cage after the rats had completed 25 out of 50 trials. For the other half of the rats, we gave empty ramekins in their home cage (sham condition). After 30 min, we placed the rats back into the operant chamber where they completed the remaining 25 trials in PLA. The next day, we gave training pellets to rats that received empty ramekins on the first day and vice versa. We ran the chow satiety test in a subset of rats (n=7) using the same experimental design as the pellet satiety test but replaced the food pellets in the ramekins with homecage chow instead.

## Fentanyl i.p injections

We injected 5 µg/kg i.p fentanyl (Cayman Chemical) or vehicle in rats (n=4) 5 min before PLA sessions. We selected this dose based on pilot experiments. In two counterbalanced PLA sessions, we gave the rats either i.p injection of fentanyl or saline.

## Fiber photometry

We used LEDs (ThorLabs) to deliver 465 nm (wavelength to excite $GRAB_{DA}$) and 405 nm (isosbestic control) and measure dopamine activity. The isosbestic signal is used as a control for fiber bleaching and motion artifacts as it is subtracted from the 465 nm signal during analysis. We sinusoidally modulated the intensity of the 465 nm and 405 nm light at 210 and 337 Hz, respectively and connected the LEDs to a four-port fluorescence mini cube (Doric Lenses). The combined LED output passed through a fiber optic cable (1 m long; 400 µm core; 0.48 NA; Doric Lenses) which was connected to the implanted fiber optics with sleeves. We maintained the light intensity at the tip of the fiber

optic cable at 10–15 µW across behavioral sessions. We collected the GRAB$_{DA}$ and isosbestic control channel emission using the same fiber optic cable and focused the emission light onto a photoreceiver (Newport). We low pass filtered and digitized the emission light at 3 Hz and 5 KHz, respectively by a digital processor controlled by Synapse software suite (RZ5P, Tucker Davis Technologies (TDT)). We time-stamped the behavioral events including lever insertion/retraction, lever press, food cup entry, etc. by sending them as TTL (transistor-transistor logic) pulses to Synapse software.

## Histology

After all behavioral testing, we deeply anesthetized rats with isoflurane and transcardially perfused them with 200 mL of 0.1 M PBS followed by 400 mL of 4% paraformaldehyde (PFA) in distilled H$_2$O. We quickly removed the brains and post-fixed them in 4% PFA for at least 2 hr before we transferred them to 30% sucrose in PBS for 48 hr at 4 °C. We subsequently froze the brains using dry ice and stored them at −20 °C until sectioning. We collected 50 µm coronal sections containing BNST on a cryostat (Leica Microsystems) and preserved them in a cryopreservant. We mounted the sections on slides and coverslipped them with Vectashield mounting medium with DAPI (Vector Laboratories). We verified fiber optic placements and viral expression in the dBNST using anatomical boundaries defined by *Paxinos and Watson, 2006* under a confocal microscope. A representative example and summary of GRAB$_{DA}$ expression and fiber placements are shown in *Figure 1C*.

## Photometry analysis

We analyzed the signals using custom-written MATLAB (Mathworks) scripts. We calculated ΔF/F (z score) by smoothing signals from the isosbestic control channel (*Lerner et al., 2015*; *Root et al., 2020*). We regressed the isosbestic signal onto the GRAB$_{DA}$-dependent signal to create a fitted isosbestic signal by using the linear model generated during the regression. We then calculated z scores by subtracting the fitted isosbestic signal from the GRAB$_{DA}$-dependent signal and dividing by the fitted isosbestic signal. This resulted in a GRAB$_{DA}$ signal devoid of artifacts created by photobleaching, fiber bending, or movements. We collected z scores in the behavioral window of interest defined as 5 s before cue onset to 10 s after pellet delivery. We quantified the area under the curve (AUC) in the 2 s following cue onset and pellet delivery and independently calculated these parameters for CS+ and CS− trials. In all dopamine signal analyses, unless otherwise noted, we subtract CS− signal from the CS+ signal. We defined significant transients in our behavioral window if the peak amplitude during the trials (0 to +20 s relative to cue onset) was 2z-score (p=0.05) above baseline (5 s prior to cue onset) during the entire behavioral window on Day 1 or Day 5 of PLA. Furthermore, to ensure these signals were time-locked to cues and not spurious, we calculated 95% confidence intervals using bootstrapped resampling (1000 resamples) of all trials' photometry data for each rat across CS+ trials of Day 5 of PLA. Most rats displayed a consistent, robust increase in the signal reaching significantly above baseline within 70 milliseconds of CS+ onset, that stayed above baseline for a minimum of 40 milliseconds, consecutively. Four rats did not meet either of these criteria (greater than 2z-score peak signal or 40 ms of consecutive time with the significantly elevated signal at CS+) and were excluded. Their data is in *Figure 2—figure supplement 3*. All included rats met both criteria. We also removed trials where the patch cord disconnected from further signal processing.

## Statistical analysis

We analyzed the data using SPSS, GraphPad Prism, and Matlab. We used mixed design repeated measures ANOVAs to analyze PLA behavioral and GRAB$_{DA}$ signal data. Whenever ANOVAs revealed significant interactions between groups, we ran t-tests with Bonferroni corrections for multiple comparisons to guard against Type I errors. We define dependent measures, within/between-subject factors, and report significant effects and interactions in the corresponding results section.

## Code availability

Modified TDT-supplied MATLAB code is available on GitHub (https://github.com/ugyawali/photometry copy archived at *Gyawali, 2022*).

## Acknowledgements

We thank Jessie Feng for technical assistance and the Animal Care Facility for colony maintenance. We thank Asaf Keller and the Department of Anatomy and Neurobiology for sharing photometry equipment. Illustrations created in Biorender.com

## Additional information

### Funding

| Funder | Grant reference number | Author |
|---|---|---|
| McKnight Foundation | | Donna Calu |
| National Institute on Drug Abuse | R01DA043533 | Donna Calu |
| University of Maryland, Baltimore | | Donna Calu |

The funders had no role in study design, data collection and interpretation, or the decision to submit the work for publication.

### Author contributions

Utsav Gyawali, Conceptualization, Data curation, Software, Formal analysis, Validation, Investigation, Visualization, Methodology, Writing - original draft, Writing – review and editing; David A Martin, Software, Formal analysis, Methodology, Writing – review and editing; Fangmiao Sun, Yulong Li, Resources; Donna Calu, Conceptualization, Resources, Formal analysis, Supervision, Funding acquisition, Writing - original draft, Project administration, Writing – review and editing

### Author ORCIDs

Utsav Gyawali http://orcid.org/0000-0001-5072-6780
Yulong Li http://orcid.org/0000-0002-9166-9919
Donna Calu http://orcid.org/0000-0003-2377-9494

### Ethics

We performed all experiments in accordance with the 'Guide for the care and use of laboratory animals' (8th edition, 2011, US National Research Council), and the University of Maryland Institutional Animal Care and Use Committee approved all experimental procedures (IACUC protocol number: 0919007).

### Decision letter and Author response

Decision letter https://doi.org/10.7554/eLife.81980.sa1
Author response https://doi.org/10.7554/eLife.81980.sa2

## Additional files

### Supplementary files

• MDAR checklist

### Data availability

The data used in this manuscript are available on Zenodo (DOI: https://doi.org/10.5281/zenodo.7947009).

The following dataset was generated:

| Author(s) | Year | Dataset title | Dataset URL | Database and Identifier |
|---|---|---|---|---|
| Gyawali U, Martin DA, Sun F, Li Y, Calu DJ | 2022 | Dopamine in the Dorsal Bed Nucleus of Stria Terminalis signals Pavlovian sign-tracking and reward violations | https://zenodo.org/record/7947009 | Zenodo, 10.5281/zenodo.7947009 |

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
