## [Editor Report]

Gyawali et al. report individual differences in extended amygdala dopamine signaling of natural and drug reward associated cues. The authors provide compelling evidence of dopamine correlates of Pavlovian natural reward and instrumental drug reward associations in rats, and their results are of broad interest to those studying brain reward systems with significance for cue-induced relapse vulnerability, in particular.

---

## [Decision Letter]

**Decision letter after peer review:**

Thank you for submitting your article "Dopamine in the Dorsal Bed Nucleus of Stria Terminalis signals Pavlovian sign-tracking and reward violations" for consideration by *eLife*. Your article has been reviewed by 3 peer reviewers, and the evaluation has been overseen by a Reviewing Editor and Michael Taffe as the Senior Editor. The reviewers have opted to remain anonymous.

Essential revisions:

1) Additional data and analyses for the fentanyl self administration experiments are needed given the low sample size. Relatedly, some explanation for exclusion of 26 rats is warranted.

2) There should be a more thorough discussion of inputs from different DA sources.

3) Analysis of shuffled vs. time-locked signal to control for spontaneous fluorescence is a minimum requirement, though ideally some pilot data with a control sensor (see individual reviewer comments).

4) Better comparison of sign-tracking vs. goal-tracking in the fentanyl experiment and tighter integration with reward prediction error (RPE) analyses.

*Reviewer #1 (Recommendations for the authors):*

This study is well conducted and provides new information as to how DA is signaling within the BNST during reward-relevant behaviors. At present though I have some questions that I think would guide/improve the study:

The big question is where is the DA coming from during these relevant behaviors. The BNST receives DA from both the VTA and the PAG but the anatomical distribution of this innervation is distinct https://pubmed.ncbi.nlm.nih.gov/26792442/. The discussion gets into this a bit, however, the lack of pathway-specific studies is an absence to fully understanding how what is being observed in the BNST is functionally distinct from what is happening with DA encoding in the NAc.

26 rats were excluded for various reasons, 2 additional for catheter patency.

The methods state that transients were excluded when they were not equal to or greater than 2 z-scores above baseline. This seems excessive and may miss smaller events? Perhaps this wording is confusing in the methods bc while the transients are over 2, the "sustained DA" as measured in AUC seems to be under 2z. Further, the transients in Figure 2B don't appear to be over 2z in the representative traces. This being said, I've recently been made aware that there are issues with autofluorescence, and blood flow that can result in optical signals that do not correspond to the detection of an analyte. The authors should demonstrate that they do not observe behaviorally relevant signals with the control GRAB DA virus. (https://www.addgene.org/140555/)

In discussing dBNST NE input, it would be good to cite the following where NE release was directly measured in the dBNST: https://pubmed.ncbi.nlm.nih.gov/20128849/ (see Figure 3).

Were there any differences in dBNST DA signaling with the fentanyl SA between the sign trackers and the goal trackers?

The authors state: "Microdialysis studies establish that several classes of drugs of abuse, including opioids, increase tonic DA in the BNST (Carboni et al., 2000)." As microdialysis cannot distinguish tonic vs. phasic dopamine signals, I would not describe this as "tonic" DA.

For discussion of phasic dopamine in the BNST with voltammetry and opioids (Line 561), this should be cited: https://www.nature.com/articles/npp2016135

The discussion on sex differences within dopaminergic signaling in the BNST would benefit from the following papers examining PAG dopamine neurons, pain, and sex differences: https://www.nature.com/articles/s41598-021-91672-8, https://pubmed.ncbi.nlm.nih.gov/33740416/

*Reviewer #2 (Recommendations for the authors):*

1) The authors interestingly find that, after intermittent fentanyl self-administration, cues associated with fentanyl reward appear to trigger enhanced BNST dopamine release. However, the small group of rats (n=3) makes it difficult to appropriately make this assertion in the same thorough manner as in earlier experiments. This interesting trend would be further strengthened with a larger group of rats that allow enough power to detect interaction in Figure 5C, for example. It would also be interesting to know whether the 3 rats tested in the fentanyl probe test were sign-tracking or goal-tracking rats.

2) How do the GRABDA expression/optic fiber placements overlap with dopaminergic fibers in dBNST? – For example, how does the density of dopaminergic terminals span dBNST, from rostral to caudal? This could potentially account for differences in the magnitude of the signal seen between rats, or between goal and sign trackers (although most of their placements look to be among mostly overlapping Bregma points).

3) In the methods section (lines 114-115), the text states that rats were excluded if there was a 'lack of significant GRABDA signal during behavioral event compared to baseline (N=26)". Please clarify what behavioral event was used. Did the experimenters use a lack of response to reward or reward CS+ in the PLA as being indicative of a low/no signal? This could be potentially important if only those rats that showed dopamine response to reward and reward cues during PLA were included and others were not. In that case, it would mean a substantial proportion of animals where dBNST dopamine release does not occur in dBNST during said tasks. In such a case, that proportion should be reported and traces depicted in Figure 1 (or supplement to Figure 1).

4) Regarding the analysis in Figure 3, I appreciate that the authors chose to show z-scores for each of the 2s time bins following reward delivery/non-delivery. This is important in light of the fact that judging from the traces, there seems to be a biphasic signal during negative reward error trials (a brief positive response followed by a negative signal). Yet, it is unclear why this analysis strategy is abandoned for the insets of Figures3G-H that show analysis that averages the signal across the entire 6s bin (as opposed to the above Figure 3D which parses each time bin). As a result, it appears that there is no effect of positive or negative RPE signals (especially in STs). Instead, in panels G-H, it might make more sense to compare z scores during each bin among STs and then among GTs. Otherwise, it's difficult to discern whether STs show any negative reward prediction signaling in dBNST. Relatedly, for the same set of analyses in Fig3G-H (also in text lines 406-412), comparing positive to negative trial z scores among each type of rat (goal- or sign- tracking) seems an odd choice – because these trials are independent and it's unclear what is gained from comparing the two. Rather, a more insightful analysis could be to see whether z-scores on positive or negative trials among each group differ significantly from zero (i.e., >2) or from z-scores on 'expected' trials.

*Reviewer #3 (Recommendations for the authors):*

I have comments on some of the analysis and interpretations of the data overall.

The RPE manipulation is a nice addition and really broadens the scope of the dopamine investigation. I'm a little confused about the approach to determining if a positive or negative RPE is signaled by the recorded GRAB fluorescence, however. Generally, it seems like the changes in dopamine based on expectation violations are temporally specific, which motivates the data binning in Figure 3 – but it's a bit unclear what statistical comparisons are significant. Maybe an area under/below the curve analysis would help this a little.

Related, the positive vs negative RPE comparisons for STs vs GT/INT is also confusing – it seems that only positive vs negative trials are contrasted statistically (Figure 3G+H). This is where the difference between tracking groups comes – with STs not differing between positive and negative, but GT/INTs showing elevated signals in positive vs negative trials. First, which part/bin/time of the signal being compared here is not clear. Second, to me in order to really say that a positive or negative RPE has been signaled the dopamine response would need to be different in positive vs expected and expected vs negative conditions. The fact that GT/INT dopamine more clearly discriminates against positive and negative expectation violations is still meaningful but it doesn't seem quite the same as "encode bidirectional RPE" without further analysis. Overall I feel like the analysis of this section could be beefed up and expanded. I also think the extension of RPE encoding questions to the BNST is the most impactful part of the data.

More or less, the dopamine signals recorded in BNST follow classic striatal/midbrain dopamine encoding. That is interesting and to me a little unexpected, given the role BNST has in stress, anxiety, and other negative states. Perhaps a little more discussion of how these signals do and do not compare to classic striatal dopamine is warranted. Also given the quite distinct dopamine signals seen in the tail of the striatum, which also come from nigra dopamine neurons (rather than raphe and another place), it is also surprising to see such "normal" RPE-related dynamics in a non-striatal region.

The satiety experiment and fentanyl results are interesting, but in the scope of the paper in the current form, they felt disconnected, especially given that the ST/GT tracking component of the investigation is not carried through. It just feels a bit like two different papers, perhaps these elements of the data could be better linked.

[Editors' note: further revisions were suggested prior to acceptance, as described below.]

Thank you for resubmitting your work entitled "Dopamine in the Dorsal Bed Nucleus of Stria Terminalis signals Pavlovian sign-tracking and reward violations" for further consideration by *eLife*. Your revised article has been evaluated by me as Senior Editor and a Reviewing Editor.

The manuscript has been improved but there are some remaining issues that need to be addressed, as outlined below:

The authors have introduced new questions about rat exclusion criteria that need to be clarified. This could be addressed by showing more of the excluded rat data since it is unclear if authors may have excluded rats that had a good signal but missed placements. This would also bolster the central RPE conclusions and specificity of the signals.

*Reviewer #1 (Recommendations for the authors):*

I am still enthusiastic about this study and the contributions it makes to the field. I appreciate the authors' careful and thoughtful reply to the reviewers; however, I am still a bit concerned about some facets of the paper and look forward to discussing this with the other reviewers in the consult session.

In particular, I am concerned that the authors did not conduct the shuffling analysis, which the reviewers requested was important. The bootstrapping method is interesting, but I am not sure if animals should be excluded based on it, perhaps the other reviewers can elaborate. In the same vein, the distinct BNST inputs (VTA and PAG) release DA very differently, therefore smaller signals may be relevant, and they are still excluding small signals. The fiber placement is too large to determine if there is input coming into the oval BNST (more PAG input) vs. the juxtacapsular (more VTA input); and, it is concerning that there were some rats with no signals at all (contributing to a large number of animals that were excluded from the study).

Finally, the removal of the fentanyl SA study, while focusing on the paper, does eliminate some of the excitement. I do hope the authors add additional N and either publish with this manuscript or in a subsequent manuscript

*Reviewer #2 (Recommendations for the authors):*

The authors here addressed all my concerns, regarding the inclusion of fentanyl self-administration, analysis of ST/GT RPE photometry, and expansion of important discussion points regarding dopamine inputs to dBNST. I also appreciate their added explanation of exclusion criteria in photometry experiments. The overall revisions help strengthen the authors' conclusion that dBNST dopamine contributes to cue-induced motivated behavior and is influenced by factors such as satiety and opioids. These findings add new and important insight to the role of dBNST in reward-related behavior beyond the more classical role of negative/stress-related motivation.

*Reviewer #3 (Recommendations for the authors):*

Thanks to the authors for this revised manuscript. My original comments have been addressed and I think the paper is stronger and more focused. I have a few remaining comments mostly about the excluded rats.

A little clarification. In the rebuttal, the authors say "We had 20 rats that were not excluded for technical reasons. Of these, 4 rats were excluded for not meeting our (revised) standards of having sufficient photometry signals (see below)." But then in the revised manuscript text, it says – "We excluded rats because of lack of viral expression (N = 4), incorrect fiber optic placements (N = 6), and headcap loss (N = 4). Additionally, we excluded rats (N = 8) that showed food cup entry artifacts before we optimized our photometry setup. The artifacts resulted in loss of signal due to the patch cord hitting the wall of the foodcup. Finally, rats (N = 4) presented in Figure 2 —figure supplement 3 were excluded that had sufficient viral expression and fiber placement but did not show robust photometry signals".

- Unless I'm misunderstanding it sounds like only 4 rats were excluded for non-technical reasons and the other 22 had various technical issues.

For the 6 rats with misplaced optic fibers – did these rats have measurable dopamine signals, or is this a matter of missing the virus expression completely? If there are rats with viable signals that happen to be outside of the BNST that would be an interesting control comparison. Alternatively, if these 6 rats have measured photometry signals but the fibers aren't above virus expression, you could include them as controls with the new confidence interval analyses in a supplement – presumably there will be no meaningful autofluorescence signals in these rats. Either way, this could potentially further strengthen the main dataset.

Did the 4 excluded rats with no cue signals develop conditioned behavior at similar levels to the main data set rats? I would suggest including the behavioral data for these rats alongside the photometry data in the corresponding supplemental figure.

8 rats were excluded because of signal artifacts associated with the port entry. It's unclear from the behavior data presented when rats are making port entries (presumably this varies substantially by tracking phenotype) but if there is substantial CS onset data that is not contaminated by port entry, that could be a meaningful addition given the large size of this exclusive group.

---

## [Author Response]

Essential revisions:1) Additional data and analyses for the fentanyl experiments are needed given the low sample size. Relatedly, some explanation for exclusion of 26 rats is warranted.

As detailed below in point by point response to reviewers, we have refocused the manuscript on sign- and goal-tracking differences during Pavlovian approach and reward violations.

We have removed the fentanyl self-administration data, which not only were conceptually extraneous, but also had low sample size. We have detailed our justification of rat exclusions and include a supplemental figure that parallels figure 2 for the rats with signals that failed to reach our statistical threshold for inclusion. We added a statement in the discussion that describes the possibility that some rats may have been excluded that had sufficient viral expression and fiber placement, but that did not show evidence of dopamine binding in the dBNST.

2) There should be a more thorough discussion of inputs from different DA sources.

We have added discussion on dopaminergic inputs to BNST.

3) Analysis of shuffled vs. time-locked signal to control for spontaneous fluorescence is a minimum requirement, though ideally some pilot data with a control sensor (see individual reviewer comments).

To address concerns regarding the strength and timing of our signals relative to cue presentation, we reanalyzed all animals with any discernible signals on Day 5 of PLA during CS+ presentations using a bootstrap analysis to generate 95% confidence intervals across time. We implemented a criterion that the signals must significantly increase above the baseline signal within 70 ms of CS+ onset and stay above baseline for a minimum of 40 ms, consecutively. This second requirement greatly reduces the possibility that a chance signal crossing above 0 is mistaken for a real signal and reduces the corresponding false positive rate. All animals included in our original analysis meet these criteria, while 4 animals were excluded (Note these animals had all previously been excluded for not reaching the 2 z-score peak height minimum) – as now stated in the Methods section and shown in Figure 2 —figure supplement 3. While we cannot address the issue of blood flow changes or other biological phenomena leading to fluorescence changes without conducting lengthy additional experiments with a control sensor, we have no reason to doubt that our signals reflect dopamine given the extensive characterization of these GRAB_DA_ sensors in other brain areas in which verified dopamine signals share very similar time courses of fluorescence changes as the signals we observe here.

4) Better comparison of sign-tracking vs. goal-tracking in the fentanyl experiment and tighter integration with reward prediction error (RPE) analyses.

We have substantially revised our RPE analysis based on reviewer 2’s thoughtful feedback. These analyses now include time bin as a factor and explicitly compare error trial types to expected reward which strengthen the conclusion that goal- but not sign-, trackers show evidence for dBNST dopamine bidirectional reward prediction error signaling.

Reviewer #1 (Recommendations for the authors):This study is well conducted and provides new information as to how DA is signaling within the BNST during reward-relevant behaviors. At present though I have some questions that I think would guide/improve the study:The big question is where is the DA coming from during these relevant behaviors. The BNST receives DA from both the VTA and the PAG but the anatomical distribution of this innervation is distinct https://pubmed.ncbi.nlm.nih.gov/26792442/. The discussion gets into this a bit, however, the lack of pathway-specific studies is an absence to fully understanding how what is being observed in the BNST is functionally distinct from what is happening with DA encoding in the NAc.

We agree this is an interesting question and something we plan to explore in the future. Ultimately, identifying the source of dopamine within the BNST during reward relevant behaviors is out of the scope of the present manuscript. However, we hope that the work presented here will spur future research on this topic. We expand on this topic in the Discussion section:

“Future studies are needed to identify the extent to which VTA and PAG/DR dopaminergic inputs contribute to the BNST DA signals observed here. Dissecting the role of each dopaminergic input in driving cue and reward related signaling and behavior will inform whether these circuits work synergistically or in competition to influence appetitive behaviors (Lin et al., 2020; Park et al., 2012, 2013). Anatomical studies indicate that a substantial proportion of putative dopaminergic projections to BNST originate in the vPAG/DR (Hasue and Shammah-Lagnado, 2002, Meloni et al., 2006). Prior work has established glutamate and dopamine co-release from the vPAG/DR projection to BNST, positioning this input to directly influence BNST synaptic plasticity and associated behaviors (Li et al., 2016). Regardless, the dopamine dynamics reported here for BNST resemble those previously reported for nucleus accumbens in related behaviors (Clark et al., 2013; Flagel et al., 2011; Hart et al., 2014; Saddoris et al., 2015, 2016), suggesting a potential role for VTA DA in shaping BNST DA signaling…” “…The tracking specific differences in BNST dopamine signaling during simple appetitive approach and reward violations observed here suggest either (1) distinct contributions of VTA and vPAG/DR to dopamine signaling observed in BNST and/or (2) individual differences in the engagement of DA systems that bias towards CSsalience or RPE processes (Chow et al., 2016, Lee et al., 2018). Consideration of tracking-specific dopamine signaling differences in future studies that employ projection-specific manipulations will aid in interpreting each projection’s contribution to BNST dopamine signaling and behavior.”

26 rats were excluded for various reasons, 2 additional for catheter patency.The methods state that transients were excluded when they were not equal to or greater than 2 z-scores above baseline. This seems excessive and may miss smaller events? Perhaps this wording is confusing in the methods bc while the transients are over 2, the "sustained DA" as measured in AUC seems to be under 2z. Further, the transients in Figure 2B don't appear to be over 2z in the representative traces. This being said, I've recently been made aware that there are issues with autofluorescence, and blood flow that can result in optical signals that do not correspond to the detection of an analyte. The authors should demonstrate that they do not observe behaviorally relevant signals with the control GRAB DA virus. (https://www.addgene.org/140555/)

We agree more detail is needed to justify rat exclusions. We have now added more robust exclusion criteria for rats that displayed viral expression and proper fiber placement, but not high levels of signal during training (4 rats) in the form of a consecutive, confidence interval test. We have now expanded the reasons in the Subjects subsection of Methods and the Photometry Analysis subsection of the Methods section:

Methods

“Subjects

…We excluded rats because of lack of viral expression (N = 4), incorrect fiber optic placements (N = 6), and headcap loss (N = 4). Additionally, we excluded rats (N = 8) that showed food cup entry artifacts before we optimized our photometry setup. The artifacts resulted in loss of signal due to the patch cord hitting the wall of the foodcup. Finally, rats (N = 4) presented in Figure 2 —figure supplement 3 were excluded that had sufficient viral expression and fiber placement but did not show robust photometry signals in the dBNST during CS+ presentation by day five of PLA training (see Photometry Analysis subsection for more details).”

“Photometry Analysis

…We defined significant transients in our behavioral window if the peak amplitude during the trials (0 to +20 s relative to cue onset) was 2 z-score (p=0.05) above baseline (5s prior to cue onset) during the entire behavioral window on Day 1 or Day 5 of PLA. Furthermore, to ensure these signals were time-locked to cues and not spurious, we calculated 95% confidence intervals using bootstrapped resampling (1000 resamples) of all trials' photometry data for each rat across CS+ trials of Day 5 of PLA. Most rats displayed a consistent, robust increase in signal reaching significantly above baseline within 70 milliseconds of CS+ onset, that stayed above baseline for a minimum of 40 milliseconds, consecutively. 4 rats did not meet either of these criteria (greater than 2 z-score peak signal or 40 ms of consecutive time with significantly elevated signal at CS+) and were excluded. Their data is in Figure 2 —figure supplement 3. All included rats met both criteria. We also removed trials where the patch cord disconnected from further signal processing.”

The addition of the consecutive-confidence-interval criteria should rule out spurious signals, and potentially autofluorescence artifacts. Some rats (N = 4) had viral expression and correct fiber placement but did not show peak amplitude 2 z-score above baseline (during the trial) on either Day 1 or Day 5 of PLA. In Figure 2—figure supplement 3, we now show parallel analysis to Figure 2 from excluded rats that do not meet the 2 z-score peak and threshold and confidence interval analyses. Based on these analyses we are confident we are not missing smaller events, as excluded rats with expression do not show robust event related activity changes. We also added a brief discussion about excluded rats that had sufficient viral expression and fiber placement, but that did not show evidence of dopamine binding in the dBNST. As noted above in this response letter, our signals display time courses consistent with well characterized dopamine signaling using this sensor in other brain areas, leading us to conclude that blood flow changes are unlikely to be the source of the signal.

In discussing dBNST NE input, it would be good to cite the following where NE release was directly measured in the dBNST: https://pubmed.ncbi.nlm.nih.gov/20128849/ (see Figure 3).

We now include this reference and add the finding about relatively slow NE clearance in the dBNST:

“Even though the GRAB_DA_ construct we used is 15-fold more sensitive to dopamine than norepinephrine (NE), BNST norepinephrine plays an important role in motivated behaviors and dBNST receives dense noradrenergic input with relatively slow NE clearance measured in vivo (Egli et al., 2005; Flavin and Winder, 2013; Park et al., 2009; Sun et al., 2020).”

Were there any differences in dBNST DA signaling with the fentanyl SA between the sign trackers and the goal trackers?

This is a very interesting question, but since our sample size for this phase of the study was low, we have now removed the fentanyl self-administration data. Instead, we now focus the manuscript on sign- and goal-tracking differences during Pavlovian approach and reward violations.

The authors state: "Microdialysis studies establish that several classes of drugs of abuse, including opioids, increase tonic DA in the BNST (Carboni et al., 2000)." As microdialysis cannot distinguish tonic vs. phasic dopamine signals, I would not describe this as "tonic" DA.

We agree with the reviewer and have changed the sentence accordingly:

“Microdialysis studies establish that several classes of drugs of abuse, including opioids, increase DA in the BNST (Carboni et al., 2000).”

For discussion of phasic dopamine in the BNST with voltammetry and opioids (Line 561), this should be cited: https://www.nature.com/articles/npp2016135

We now add the citation the reviewer suggests.

The discussion on sex differences within dopaminergic signaling in the BNST would benefit from the following papers examining PAG dopamine neurons, pain, and sex differences: https://www.nature.com/articles/s41598-021-91672-8, https://pubmed.ncbi.nlm.nih.gov/33740416/

We agree with the reviewer and as such have edited existing text and added a brief discussion based on the findings from these studies.

“The BNST is a sexually dimorphic brain region (Hisasue et al., 2010; Shah et al., 2004; Tsuneoka et al., 2017), highlighting the necessity of studying both sexes to fully understand the contribution of BNST DA to motivated behavior. Dopaminergic projections from vPAG/DR-BNST play sexspecific roles, with pathway activation associated with distinct pain and locomotor behavioral changes for males and females, respectively (Yu et al., 2021). We used both male and female rats in the present study and analyzed our BNST DA photometry data from Pavlovian autoshaping, RPE, satiety test sessions using Sex instead of Tracking as a factor. While we observed no sex effects here, prior studies establish BNST-mediated sex differences in pain and locomotor behaviors as well as in opioid withdrawal (Luster et al., 2020; Yu et al. 2021a; Yu et al., 2021b).”

Reviewer #2 (Recommendations for the authors):1) The authors interestingly find that, after intermittent fentanyl self-administration, cues associated with fentanyl reward appear to trigger enhanced BNST dopamine release. However, the small group of rats (n=3) makes it difficult to appropriately make this assertion in the same thorough manner as in earlier experiments. This interesting trend would be further strengthened with a larger group of rats that allow enough power to detect interaction in Figure 5C, for example. It would also be interesting to know whether the 3 rats tested in the fentanyl probe test were sign-tracking or goal-tracking rats.

The three rats tested were N = 2 intermediate (mix of sign- and goal-tracking approach) and N = 1 sign-tracker. The self-administration phase of the experiment was not a primary aim of the study, though we agree the data are interesting. Only one cohort of rats was trained in fentanyl selfadministration, and unfortunately, we were not sufficiently powered to look within tracking groups. Conceptually, this departs from our aim to characterize individual differences in extended amygdala dopamine dynamics. Furthermore, we do not currently have the expertise (first author graduated) or resources to replicate this phase of the study, so we have removed data from the self-administration phase of the study from the manuscript. We look forward to pursuing these questions in future studies.

2) How do the GRABDA expression/optic fiber placements overlap with dopaminergic fibers in dBNST? – For example, how does the density of dopaminergic terminals span dBNST, from rostral to caudal? This could potentially account for differences in the magnitude of the signal seen between rats, or between goal and sign trackers (although most of their placements look to be among mostly overlapping Bregma points).

As the reviewer keenly points out, our placements were largely consistent (~73% (N=11) at level of bregma and only ~13% (N=2) slightly anterior(+0.12) and ~20 % (N=3) slightly posterior (-0.12) to bregma), limiting our ability to analyze the data with sufficient power to determine whether variability in expression and/or fiber placement contributed to differences in the signal along the anterior to posterior axis of BNST. Regardless, we think this is an important consideration and have consulted anatomical and functional studies (1 relevant case, no BNST coordinates given, estimated target– based on lateral ventricle, fornix and anterior commissure landmarks in Hasue and Shammah-Lagnado, 2002: Figure 7B– around 0.0, bregma, in rat), Meloni et al., 2006 (stereotaxic target -0.2 posterior to bregma in rat), Yu et al., 2021 (stereotaxic target +0.23 anterior to bregma in mouse). From these studies that demonstrate DA input anatomically and/or functionally that spans anterior (Yu et al., 2021) to posterior (Meloni et al., 2006) of our reported cases mostly around bregma (ie: Hasue and Shammah-Lagnado, 2002), we conclude there is substantial vPAG/DR DA input and to a lesser extent VTA input at the level of BNST that we measure GRABDA signals. We appreciate any further insight the reviewer may have for anatomical or functional studies that increase resolution on rostral-caudal distribution of dopaminergic inputs to BNST. We have added the following sentences to the discussion to address this important methodological limitation of our approach:

“A methodological limitation of the current approach is that variations in expression of fluorescent sensor and/or fiber placement along a gradient of DA input to BNST could potentially influence the magnitude of GRAB_DA_ measurements. Our fiber placements were largely consistent (~73% at level of bregma) and overlapped with the densest area of viral expression of the fluorescent sensor. At this level of BNST that we measure the majority of GRAB_DA_ signals, there is heavy vPAG/DR DA input and to a lesser extent VTA input (Hasue and Shammah-Lagnado, 2002). Other anatomical and/or functional studies that target BNST up to 0.2 mm anterior or posterior to bregma also observe substantial putative dopaminergic input from vPAG/DR (Meloni et al., 2006, Yu et al., 2021).”

3) In the methods section (lines 114-115), the text states that rats were excluded if there was a 'lack of significant GRABDA signal during behavioral event compared to baseline (N=26)". Please clarify what behavioral event was used. Did the experimenters use a lack of response to reward or reward CS+ in the PLA as being indicative of a low/no signal? This could be potentially important if only those rats that showed dopamine response to reward and reward cues during PLA were included and others were not. In that case, it would mean a substantial proportion of animals where dBNST dopamine release does not occur in dBNST during said tasks. In such a case, that proportion should be reported and traces depicted in Figure 1 (or supplement to Figure 1).

We regret that we did not list all the reasons for animal exclusions. We have now expanded the reasons for exclusions in the *Subjects* and *Photometry Analysis* subsection of the Methods section. These changes are also detailed above in response to Question 2 by Reviewer #1.

The reviewer’s concerns are valid regarding exclusions of animals without a dopamine response. 4 rats had viral expression and correct fiber placement but did not meet our (revised) standards of having sufficient photometry signals. In Figure 2—figure supplement 3, we now show parallel analysis to Figure 2 including signals from excluded rats that do not meet the 2 z-score threshold and confidence interval analyses. Based on these analyses we are confident we are not missing smaller events, as excluded rats with expression do not show event related activity changes. We had 20 rats that were *not* excluded for technical reasons. Of these, 4 rats were excluded for not meeting our (revised) standards of having sufficient photometry signals (see below). As we do not know why these animals did not show sufficient signal, we hesitate to speculate that this lack of signal really does reflect low dopamine in these animals, as it may be due to unidentified technical issues instead. Nevertheless, the reviewer makes a valid point, as the lack of signal may be indeed biological. We have added this point and the ratio of the animals kept vs discarded (4/20 = 0.2) in the discussion.

Methods

“Subjects

...We excluded rats because of lack of viral expression (N = 4), incorrect fiber optic placements (N = 6), and headcap loss (N = 4). Additionally, we excluded rats (N = 8) that showed food cup entry artifacts before we optimized our photometry setup. The artifacts resulted in loss of signal due to the patch cord hitting the wall of the foodcup. Finally, rats (N = 4) presented in Figure 2 —figure supplement 3 were excluded that had sufficient viral expression and fiber placement but did not show robust photometry signals in the dBNST during CS+ presentation by day five of PLA training (see Photometry Analysis subsection for more details).”

“Photometry Analysis

…We defined significant transients in our behavioral window if the peak amplitude during the trials (0 to +20 s relative to cue onset) was 2 z-score (p=0.05) above baseline (5s prior to cue onset) during the entire behavioral window on Day 1 or Day 5 of PLA. Furthermore, to ensure these signals were time-locked to cues and not spurious, we calculated 95% confidence intervals using bootstrapped resampling (1000 resamples) of all trials' photometry data for each rat across CS+ trials of Day 5 of PLA. Most rats displayed a consistent, robust increase in signal reaching significantly above baseline within 70 milliseconds of CS+ onset, that stayed above baseline for a minimum of 40 milliseconds, consecutively. 4 rats did not meet either of these criteria (greater than 2 z-score peak signal or 40 ms of consecutive time with significantly elevated signal at CS+) and were excluded. Their data is in Figure 2 —figure supplement 3. All included rats met both criteria. We also removed trials where the patch cord disconnected from further signal processing.”

Discussion

“…Regardless, some rats presented in Figure 2 —figure supplement 3 were excluded that had sufficient viral expression and fiber placement, but that did not show evidence of dopamine binding in the dBNST during these Pavlovian tasks. While we are limited from drawing conclusions from negative data, such individual differences in extended amygdala dopamine signaling may be important for interpreting differences in appetitive behaviors.”

4) Regarding the analysis in Figure 3, I appreciate that the authors chose to show z-scores for each of the 2s time bins following reward delivery/non-delivery. This is important in light of the fact that judging from the traces, there seems to be a biphasic signal during negative reward error trials (a brief positive response followed by a negative signal). Yet, it is unclear why this analysis strategy is abandoned for the insets of Figures3G-H that show analysis that averages the signal across the entire 6s bin (as opposed to the above Figure 3D which parses each time bin). As a result, it appears that there is no effect of positive or negative RPE signals (especially in STs). Instead, in panels G-H, it might make more sense to compare z scores during each bin among STs and then among GTs. Otherwise, it's difficult to discern whether STs show any negative reward prediction signaling in dBNST. Relatedly, for the same set of analyses in Fig3G-H (also in text lines 406-412), comparing positive to negative trial z scores among each type of rat (goal- or sign- tracking) seems an odd choice – because these trials are independent and it's unclear what is gained from comparing the two. Rather, a more insightful analysis could be to see whether z-scores on positive or negative trials among each group differ significantly from zero (i.e., >2) or from z-scores on 'expected' trials.

We thank the reviewer for pointing out this analysis shortfall. We have conducted suggested analyses which appear on the manuscript and below:

“First, to determine whether BNST GRAB_DA_ signals encode bidirectional reward prediction error (RPE), we compare signals on expected, positive and negative trials. Notably, because lever retraction occurs simultaneously with reward delivery, and sign- and goal-trackers may be in different locations at this time, we examine the signals during the six seconds (three 2-s bins) after reward delivery or omission, which captures the period corresponding to violations in reward expectations (Figure 3D). We performed a repeated measures ANOVA on z scores during the RPE session including Trial Type (Expected, Positive, Negative) and Bin (three 2 s bins (0-2 s, 2-4 s, 4-6 s)) as factors. We observed a difference in dBNST GRAB_DA_ signaling between the three trial types in the bins following reward delivery/omission (Figure 3D, Bin: F_(2,72)_ = 13.65, p<0.001, Bin x Trial Type: F_(4,72)_ = 13.99, p<0.001, Trial Type: F_(2,36)_ = 3.49, p=0.041). Post-hocs confirm that in the second 2-second bin (ie. 2-4 s) after reward delivery/omission, BNST GRAB_DA_ signals differed from one another for all three trial types, Expected vs. Positive (population traces in Figure 3E; p=0.013), Expected vs. Negative (population traces in Figure 3F; p=0.043) and Positive vs. Negative (p=0.0004). Across all rats, we observe that dBNST GRAB_DA_ signals reflect bidirectional reward prediction errors.

Then to determine whether there are tracking differences in dBNST RPE signals, we separately analyzed the z scores during RPE sessions in the two tracking groups. Again we examine how GRAB_DA_ signaling differs for the three trial types (expected, positive, negative) during the three 2second bins after reward delivery (population traces for STs and GT/INTs Figure 3G-H). In GT/INT rats we observed main effects of Trial (F(2,12) = 8.2, p = 0.006) and Bin (F(2,12) = 4.9, p = 0.027) and a Trial x Bin interaction (F(4,24) = 25.7, p <0.001). GT/INT rats showed evidence for both positive RPE (Trial (Expected, Positive) x Bin interaction F(2,12) = 14.5, p = 0.001) and negative RPE Trial (Expected, Negative) x Bin interaction (F(2,12) = 9.9, p = 0.003; Figure 3G inset). In GT/INT rats, we next examined the time course and found dBNST GRAB_DA_ signaling on both positive and negative trials differs from expected trials during the second 2-second bin (ie 2-4 s) after lever retraction/pellet delivery/omission (positive vs. expected p = 0.04, negative vs. expected p = 0.021). This suggests GT/INT rats show evidence for dBNST GRAB_DA_ bidirectional RPE signaling.

In a parallel analysis in ST rats considering all trial types (Expected, Positive, Negative) we also observed main effect of Bin (F(2,10) = 10.4, p = 0.004) and Trial x Bin interaction (F(4,20) = 5.4, p = 0.004). ST rats showed evidence for positive RPE (Trial (expected, positive) x Bin interaction F(2,10) 6.8, p = 0.014) but not negative RPE (Trial (expected, negative) x Bin interaction F(2,10) = 2.8, p = 0.153, Figure 3H inset). Post-hoc analyses in ST rats on the time-course failed to identify which bin GRABDA signals distinguished by trial type, however a planned analysis on the relevant second 2-s bin (ie. 2-4 s after lever retraction/pellet delivery/omission) indicates a main effect (F(2,10) = 5.3, p = 0.027) is marginally driven by Expected vs. Positive trial types (F(1,5)=5.0, p = 0.075) and not Expected vs. Negative trial types (F(1,5) = 2.0, p=0.216). This analysis suggests ST rats fail to show evidence for dBNST GRAB_DA_ bidirectional RPE signaling.”

Reviewer #3 (Recommendations for the authors):I have comments on some of the analysis and interpretations of the data overall.The RPE manipulation is a nice addition and really broadens the scope of the dopamine investigation. I'm a little confused about the approach to determining if a positive or negative RPE is signaled by the recorded GRAB fluorescence, however. Generally, it seems like the changes in dopamine based on expectation violations are temporally specific, which motivates the data binning in Figure 3 – but it's a bit unclear what statistical comparisons are significant. Maybe an area under/below the curve analysis would help this a little.

We thank the reviewer for pointing out the ambiguity that previously existed for this analysis and associated Figure 3. We have revised both the figure and the text to clarify the updated statistical analysis for the RPE sessions. The entire section of revised text appears in manuscript and also in response to the final point of reviewer 2, just above. We paste subsections of this revised text in the next point to address the specific issues outlined in the point below.

Related, the positive vs negative RPE comparisons for STs vs GT/INT is also confusing – it seems that only positive vs negative trials are contrasted statistically (Figure 3G+H). This is where the difference between tracking groups comes – with STs not differing between positive and negative, but GT/INTs showing elevated signals in positive vs negative trials. First, which part/bin/time of the signal being compared here is not clear. Second, to me in order to really say that a positive or negative RPE has been signaled the dopamine response would need to be different in positive vs expected and expected vs negative conditions. The fact that GT/INT dopamine more clearly discriminates against positive and negative expectation violations is still meaningful but it doesn't seem quite the same as "encode bidirectional RPE" without further analysis. Overall I feel like the analysis of this section could be beefed up and expanded. I also think the extension of RPE encoding questions to the BNST is the most impactful part of the data.

We thank the reviewer for the constructive feedback and positive assessment of the work. We have addressed these comments and performed repeated measures ANOVA on the timecourse of the RPE signals in ST and GT/INT rats. Below is a sub-section of the revised text accompanying Figure 3.

“In GT/INT rats we observed main effects of Trial (F_(2,12)_ = 8.2, p = 0.006) and Bin (F(2,12) = 4.9, p = 0.027) and a Trial x Bin interaction (F_(4,24)_ = 25.7, p <0.001). GT/INT rats showed evidence for both positive RPE (Trial (Expected, Positive) x Bin interaction F_(2,12)_ = 14.5, p = 0.001) and negative RPE Trial (Expected, Negative) x Bin interaction (F(2,12) = 9.9, p = 0.003; Figure 3G inset). In GT/INT rats, we next examined the time course and found dBNST GRAB_DA_ signaling on both positive and negative trials differs from expected trials during the second 2-second bin (ie 2-4 s) after lever retraction/pellet delivery/omission (positive vs. expected p = 0.04, negative vs. expected p = 0.021). This suggests GT/INT rats show evidence for dBNST GRAB_DA_ bidirectional RPE signaling.

In a parallel analysis in ST rats considering all trial types (Expected, Positive, Negative) we also observed main effect of Bin (F_(2,10)_ = 10.4, p = 0.004) and Trial x Bin interaction (F_(4,20)_ = 5.4, p = 0.004). ST rats showed evidence for positive RPE (Trial (expected, positive) x Bin interaction F_(2,10)_ = 6.8, p = 0.014) but not negative RPE (Trial (expected, negative) x Bin interaction F_(2,10)_ = 2.8, p = 0.153, Figure 3H inset). Post-hoc analyses in ST rats on the time-course failed to identify which bin GRABDA signals distinguished by trial type, however a planned analysis on the relevant second 2-s bin (ie. 2-4 s after lever retraction/pellet delivery/omission) indicates a main effect (F_(2,10)_ = 5.3, p = 0.027) is marginally driven by Expected vs. Positive trial types (F_(1,5)_=5.0, p = 0.075) and not Expected vs. Negative trial types (F_(1,5)_= 2.0, p=0.216). This analysis suggests ST rats fail to show evidence for dBNST GRAB_DA_ bidirectional RPE signaling.”

More or less, the dopamine signals recorded in BNST follow classic striatal/midbrain dopamine encoding. That is interesting and to me a little unexpected, given the role BNST has in stress, anxiety, and other negative states. Perhaps a little more discussion of how these signals do and do not compare to classic striatal dopamine is warranted. Also given the quite distinct dopamine signals seen in the tail of the striatum, which also come from nigra dopamine neurons (rather than raphe and another place), it is also surprising to see such "normal" RPE-related dynamics in a non-striatal region.

We agree and speculate a bit in the discussion about how individual differences in BNST DA signaling may arise from differences in the relative contribution of distinct inputs (VTA-BNST in bidirectional RPE seen in GT/INT group, and potentially vPAG-BNST involvement in strong CS incentive salience encoding of ST group.) Unlike the tail of the striatum, BNST receives relatively little DA input from substantia nigra (Hasue and Shammah-Lagnado, 2002; Meloni et al., 2006). In addition to the prior text briefly touching on this matter, we have added the following to the discussion.

“Regardless, the dopamine dynamics reported here for BNST resemble those previously reported for nucleus accumbens in related behaviors (Clark et al., 2013; Flagel et al., 2011; Hart et al., 2014; Saddoris et al., 2015, 2016), suggesting a potential role for VTA DA in shaping BNST DA signaling. Notably, NAc DA also shows greater CS-evoked, and a greater shift from US to CS- evoked DA in ST compared to GT (Flagel et al., 2011). To our knowledge, tracking-related differences in bidirectional RPE signaling in the NAc have not been systematically tested, however the bidirectional error encoding we observe across all rats is consistent with prior NAc voltammetry study (Hart et al., 2014). Here we report that GT/INT, but not ST, show evidence for bidirectional RPE DA signaling in the BNST. Whether this is also the case for NAc DA signaling remains an open question. Consistent with our findings, short inter-trial-intervals (ITI, similar to what we employ here) during autoshaping promote both classic NAc DA RPE signaling and goaltracking, whereas longer ITIs promote NAc DA CS-salience signaling and sign-tracking (Lee et al., 2018). Pharmacology studies show D1 receptors and NAc DA signaling drive CS-salience in sign-trackers (Chow et al., 2016; Saunders and Robinson 2012). The potentiating effects of hunger and systemic fentanyl injections on BNST DA signals observed here are in line with effects observed for NAc DA (Bassareo et al., 2013; Castro and Berridge 2014; Cone et al., 2014; Mahler et al., 2007; Pecina and Berridge 2005; Wilson et al., 1995). Notably, the NAc primarily receives input from the VTA, whereas the BNST receives DA inputs from VTA and vPAG/DR. The tracking specific differences in BNST dopamine signaling during simple appetitive approach and reward violations observed here suggest either (1) distinct contributions of VTA and vPAG/DR to dopamine signaling observed in BNST and/or (2) individual differences in the engagement of DA systems that bias towards CS-salience or RPE processes (Chow et al., 2016; Lee et al., 2018). Consideration of tracking-specific dopamine signaling differences in future studies that employ projection-specific manipulations will aid in interpreting each projection’s contribution to BNST dopamine signaling and behavior.”

The satiety experiment and fentanyl results are interesting, but in the scope of the paper in the current form, they felt disconnected, especially given that the ST/GT tracking component of the investigation is not carried through. It just feels a bit like two different papers, perhaps these elements of the data could be better linked.

We agree with the reviewer, particularly about the self-administration results, which were conceptually inconsistent with our goal to characterize individual differences in BNST DA signaling during Pavlovian behaviors. The self-administration phase of the experiment was not a primary aim of the study, though we agree the data are interesting. Only one cohort of rats was trained in fentanyl self-administration, and unfortunately, we were not sufficiently powered to look within tracking groups. Furthermore, we do not currently have the expertise (first author graduated) or resources to replicate this phase of the study, so we have removed data from the self-administration phase of the study from the manuscript. We look forward to pursuing these questions in future studies.

Given we have refocused the manuscript on BNST DA signaling during natural reward seeking, we revise the manuscript to better incorporate the satiety and i.p. fentanyl data. We feel these experiments are in line with our goal to characterize BNST DA signaling in natural reward seeking. While we do not observe tracking differences in the satiety data, we are cautious to interpret the lack of main effects or interactions with Tracking factor, due to the smaller sample size (pellet satiety N = 11, ST N = 4 , GT/INT N = 7 and chow satiety N = 7, ST N = 3 , GT/INT N = 4) in this phase of the study. We now report these N’s in the revised text. We have also added more rationale and associated citations to better incorporate these findings to what is known for the striatal dopamine system. A few excerpts are included below.

“In the current and following sections, we report the number of ST and GT/INT rats for each experimental phase but do not report tracking differences due to decreased statistical power to detect group differences. Prior studies indicate that the midbrain and striatal dopamine system tracks motivational state through satiety-dependent changes in the magnitude of dopamine responses (Cone et al., 2014; Hsu et al., 2018; Wilson et al., 1995). Here we determined whether motivational state also decreases task-related BNST GRABDA signals during lever autoshaping.”

“These results further bolster our finding that BNST GRAB_DA_ signals encode cue-outcome associations, which similar to striatal dopamine signaling, is blunted when the animal has reduced motivational drive (Cone et al., 2014; Wilson et al., 1995).“

“Opioids potentiate NAc activity and NAc DA responses to natural rewards and natural reward associated cues (Bassareo et al., 2013; Castro and Berridge 2014; Mahler et al., 2007; Pecina and Berridge 2005). Here we sought to determine whether opioids also potentiate task-related BNST GRAB_DA_ signals during natural reward seeking in lever autoshaping.”

[Editors' note: further revisions were suggested prior to acceptance, as described below.]

Reviewer #1 (Recommendations for the authors):I am still enthusiastic about this study and the contributions it makes to the field. I appreciate the authors' careful and thoughtful reply to the reviewers; however, I am still a bit concerned about some facets of the paper and look forward to discussing this with the other reviewers in the consult session.In particular, I am concerned that the authors did not conduct the shuffling analysis, which the reviewers requested was important. The bootstrapping method is interesting, but I am not sure if animals should be excluded based on it, perhaps the other reviewers can elaborate.

We have conducted the shuffling analysis, as we interpreted the editor’s request, the result of which is shown below. In this time shuffled analysis, for each trial we randomized actual z-scored dBNSTs GRABDA signal values, such that the values occur at random times. We then average each rats’ CS+ Trials across the session. Then we averaged across rats for the group level (GT and ST) analysis. Displayed below are the average curves (real (blue) vs. shuffled (red) CS+ data) across the groups (Shaded is the SEM) for the first (left column) fifth (right column) session of Pavlovian Lever autoshaping. The result of the time shuffled data analysis is the expected increase in baseline signal, with no event-related fluctuations. We do not feel this analysis adds substantive information, and include it as Author response image 1.

**Author response image 1. sa2fig1:** 

Only 4 rats were excluded for non-technical reasons (sufficient viral expression and fiber placement but did not show robust photometry signals). These rats’ data is presented in Figure 2 —figure supplement 3. We are not suggesting that smaller GRABDA signals are irrelevant. However, because smaller events are more likely to be contaminated by noise (ie. autofluorescence), we apply a standard approach in signal analysis of examining signals in our behavioral window that are significantly different from baseline (p<0.05 or z>2). Our additional confidence interval analysis further supports our approach. We have added a couple of sentences in the Discussion section to highlight this caveat:“A methodological limitation of the current approach … In addition, we analyzed signals that were significantly different from baseline (greater than 2z scores) in our behavioral window. We might have missed some behaviorally relevant signals due to this restriction. Future studies with control GRABDA virus are needed to determine how large a signal can be expected from artefactual sources (blood flow, autofluorescence, movement, etc).”

In the same vein, the distinct BNST inputs (VTA and PAG) release DA very differently, therefore smaller signals may be relevant, and they are still excluding small signals. The fiber placement is too large to determine if there is input coming into the oval BNST (more PAG input) vs. the juxtacapsular (more VTA input); and, it is concerning that there were some rats with no signals at all (contributing to a large number of animals that were excluded from the study).

We agree with the reviewer that distinct BNST inputs (VTA *vs* PAG) release dopamine differently in the BNST. Indeed, the fiber placement area is too large to determine the source of dopamine. Our goal of the manuscript was not to determine the source of dopamine, nor make any claim on what the source of dopamine signals are. Instead, our study points to the general role of BNST dopamine in response to reward associated cues and reward prediction error in sign and goal tracking rats. Here are snippets from Discussion that highlight the need for future investigation related to this point:

“Future studies are needed to identify the extent to which VTA and PAG/DR dopaminergic inputs contribute to the BNST DA signals observed here. Dissecting the role of each dopaminergic input in driving cue and reward related signaling and behavior will inform whether these circuits work synergistically or in competition to influence appetitive behaviors (Lin et al., 2020; Park et al., 2012, 2013)…”

“… The tracking specific differences in BNST dopamine signaling during simple appetitive approach and reward violations observed here suggest either (1) distinct contributions of VTA and vPAG/DR to dopamine signaling observed in BNST and/or (2) individual differences in the engagement of DA systems that bias towards CS-salience or RPE processes (Chow et al., 2016; Lee et al., 2018). Consideration of tracking-specific dopamine signaling differences in future studies that employ projection-specific manipulations will aid in interpreting each projection’s contribution to BNST dopamine signaling and behavior.”

Finally, the removal of the fentanyl SA study, while focusing on the paper, does eliminate some of the excitement. I do hope the authors add additional N and either publish with this manuscript or in a subsequent manuscript

We thank the reviewer for their interest in this future work extending our findings to drug reward signaling.

Reviewer #3 (Recommendations for the authors):Thanks to the authors for this revised manuscript. My original comments have been addressed and I think the paper is stronger and more focused. I have a few remaining comments mostly about the excluded rats.A little clarification. In the rebuttal, the authors say "We had 20 rats that were not excluded for technical reasons. Of these, 4 rats were excluded for not meeting our (revised) standards of having sufficient photometry signals (see below)." But then in the revised manuscript text, it says – "We excluded rats because of lack of viral expression (N = 4), incorrect fiber optic placements (N = 6), and headcap loss (N = 4). Additionally, we excluded rats (N = 8) that showed food cup entry artifacts before we optimized our photometry setup. The artifacts resulted in loss of signal due to the patch cord hitting the wall of the foodcup. Finally, rats (N = 4) presented in Figure 2 —figure supplement 3 were excluded that had sufficient viral expression and fiber placement but did not show robust photometry signals".– Unless I'm misunderstanding it sounds like only 4 rats were excluded for non-technical reasons and the other 22 had various technical issues.

The reviewer is correct. 4 rats were excluded for non-technical reasons and the other 22 had various technical issues. This was a typo in the rebuttal. We now also include behavior data from those 4 rats in Figure 2 —figure supplement 3.

For the 6 rats with misplaced optic fibers – did these rats have measurable dopamine signals, or is this a matter of missing the virus expression completely? If there are rats with viable signals that happen to be outside of the BNST that would be an interesting control comparison. Alternatively, if these 6 rats have measured photometry signals but the fibers aren't above virus expression, you could include them as controls with the new confidence interval analyses in a supplement – presumably there will be no meaningful autofluorescence signals in these rats. Either way, this could potentially further strengthen the main dataset.

All 6 rats had dBNST viral expression but misplaced fiber placements outside of the dBNST. Most of these rats were run for 3-5 sessions before they were no longer imaged due to lack of real time signal and limited photometry setups. Regardless, we analyzed the fluorescence signals from the last recording session from these rats (session 3-5 depending on when they were no longer run) and qualitatively found no consistent fluctuations in signal in the behavioral window. We also performed a confidence interval analysis on these 6 rats. Only 1/6 of these rats with misplaced cannula exhibited a signal significantly and consistently above baseline following CS+ onset (according to the parameters of our bootstrapping confidence interval test described in the methods) on the last day of training, however, the average maximum of this signal was very small, and did not exceed 2z above baseline. Therefore, all six rats with misplaced cannula had little to no appreciable signal.

Did the 4 excluded rats with no cue signals develop conditioned behavior at similar levels to the main data set rats? I would suggest including the behavioral data for these rats alongside the photometry data in the corresponding supplemental figure.

Yes, these rats developed conditioned behavior similar to the main data set rats. The data are now added in Figure 2 —figure supplement 3.

8 rats were excluded because of signal artifacts associated with the port entry. It's unclear from the behavior data presented when rats are making port entries (presumably this varies substantially by tracking phenotype) but if there is substantial CS onset data that is not contaminated by port entry, that could be a meaningful addition given the large size of this exclusive group.

Out of the 8 rats, there were 2 STs and 6 GT/INTs. The population photometry signals and associated behavior from these rats are now added as a supplement in Figure 2 —figure supplement 4. Since we also do unconditioned stimulus (US) analyses, we exclude these rats completely from the study since we do not have reliable signal from the time of US delivery for these rats. However, qualitatively we observe that these rats show similar patterns of CS-evoked dBNST dopamine signals and behavior relative to rats included in the reported data set (Figure 2) that have the continuous uncontaminated signals across the trial.